



# First ground-based FTIR observations of HFC-23 at Rikubetsu, Japan, and Syowa Station, Antarctica

Masanori Takeda[1, 2], Hideaki Nakajima[2, 1], Isao Murata[1], Tomoo Nagahama[3], Isamu Morino[2], Geoffrey C. Toon[4], Ray F. Weiss[5], Jens Mühle[5], Paul B. Krummel[6], Paul J. Fraser[6], and Hsiang-Jui Wang[7]

[1]Graduate School of Environmental Studies, Tohoku University, Sendai, Miyagi, 980-8572, Japan
[2]National Institute for Environmental Studies, Tsukuba, Ibaraki, 305-8506, Japan
[3]Institute for Space-Earth Environmental Research, Nagoya University, Nagoya, Aichi, 464-8601, Japan
[4]Jet Propulsion Laboratory, California Institute of Technology, Pasadena, CA, 91109, U. S. A.
[5]Scripps Institution of Oceanography (SIO), University of California San Diego, La Jolla, CA, 92093-0244, U. S. A.
[6]Climate Science Centre, Commonwealth Scientific and Industrial Research Organisation (CSIRO), Oceans and Atmosphere, Aspendale, Victoria, 3195, Australia
[7]School of Earth and Atmospheric Sciences, Georgia Institute of Technology, Atlanta, GA, 30332-0340, U. S. A.

*Correspondence to*: Masanori Takeda (takeda.masanori@nies.go.jp)

**Abstract.**

We have developed a procedure for retrieving atmospheric abundances of HFC-23 ($CHF_3$) with a ground-based Fourier transform infrared spectrometer (FTIR) and analysed the spectra observed at Rikubetsu, Japan (43.5ºN, 143.8ºE), and at Syowa Station, Antarctica (69.0ºS, 39.6ºE). The FTIR retrievals were carried out with the SFIT4 retrieval program, and the two spectral windows of 1138.5–1148.0 $cm^{-1}$ and 1154.0–1160.0 $cm^{-1}$ in the overlapping $\nu_2$ and $\nu_5$ vibrational-rotational transition bands of HFC-23 were used to avoid strong $H_2O$ absorption features. We considered $O_3$, $N_2O$, $CH_4$, $H_2O$, HDO, CFC-12 ($CCl_2F_2$), HCFC-22 ($CHClF_2$), PAN ($CH_3C(O)OONO_2$), HCFC-141b ($CH_3CCl_2F$), and HCFC-142b ($CH_3CClF_2$) as interfering species. Vertical profiles of $H_2O$, HDO, and $CH_4$ are preliminarily retrieved with other independent spectral windows because these profiles may induce large uncertainties in the HFC-23 retrieval. Each HFC-23 retrieval has only one piece of vertical information with sensitivity to HFC-23 in the troposphere and the lower stratosphere. The retrieval errors mainly arise from the systematic uncertainties of the spectroscopic parameters used to obtain the HFC-23, $H_2O$, HDO, and $CH_4$ abundances. For comparison between FTIR-retrieved HFC-23 total columns and surface dry-air mole fractions provided by AGAGE (Advanced Global Atmospheric Gases Experiment), the FTIR-retrieved HFC-23 dry-air column-averaged mole fractions ($X_{HFC-23}$) were calculated. The FTIR-retrieved $X_{HFC-23}$ at Rikubetsu and Syowa Station have negative biases compared to AGAGE datasets. The trend derived from the FTIR-retrieved $X_{HFC-23}$ data at Rikubetsu for December to February (DJF) data over the 1997–2010 period is $0.817 \pm 0.087$ ppt (parts per trillion) year$^{-1}$, which is in good agreement with the trend derived from the annual global mean datasets of the AGAGE 12-box model for the same period ($0.820 \pm 0.011$ ppt year$^{-1}$). The trend of the FTIR-retrieved $X_{HFC-23}$ data at Rikubetsu for DJF data over the 2007–2020 period is $0.894 \pm 0.099$ ppt year$^{-1}$, which is smaller than



the trend in the AGAGE in-situ measurements at Trinidad Head (41.1ºN, 124.2ºW) for the 2007–2019 period (0.984 ± 0.002 ppt year$^{-1}$). The trend computed from the $X_{HFC-23}$ datasets at Syowa Station over the 2007–2016 period is 0.823 ± 0.075 ppt year$^{-1}$, which is consistent with that derived from the AGAGE in-situ measurements at Cape Grim (40.7ºS, 144.7ºE) for the same period (0.874 ± 0.002 ppt year$^{-1}$). Although there are systematic biases on the FTIR-retrieved $X_{HFC-23}$ at both sites, these

results indicate that ground-based FTIR observations have the capability to monitor the trend of atmospheric HFC-23.

## 1. Introduction

Trifluoromethane (CHF$_3$), also known as hydrofluorocarbon-23 (HFC-23), has an atmospheric lifetime of 228 years and a global warming potential integrated over a 100-year time scale (100-year GWP) of 12,690 (Montzka et al., 2019). Due to this high GWP, emissions of HFC-23 are contributing to climate change. HFC-23 is an unwanted by-product of the production of

chlorodifluoromethane (CHClF$_2$), hydrochlorofluorocarbon-22 (HCFC-22), with the HFC-23/HCFC-22 production ratio estimated to be up to 4% (McCulloch and Lindley, 2007).

Under the regulations of the "Montreal Protocol on Substances that Deplete the Ozone Layer (Montreal Protocol)" (UNEP, 2000), production and consumption of ozone-destroying chlorofluorocarbons (CFCs) have been completely banned since 2010, whereas production and consumption of hydrochlorofluorocarbons (HCFCs), which have less effect on ozone depletion, have

continued. The Montreal Protocol is phasing out the production and consumption of HCFCs for emissive uses by 2020 in developed countries, and by 2030 in developing countries, while use for feedstock (e.g. in production of HFCs and fluoropolymers) is not restricted. Hence, emissions of HCFCs to the earth's atmosphere are expected to continue for quite a while. HCFC-22, one of the major HCFCs with an ozone depletion potential (ODP) of ~0.03 and a 100-year GWP of 1,760 (Harris et al., 2014), has been widely used in air conditioners, refrigerators, foaming agents, or heat insulating materials, and

therefore large banks still exist, which also contribute to ongoing emissions. Emission of HCFC-22 has increased since 2004 (Montzka et al., 2009), and global emissions in 2010 are estimated to have reached 386 ± 41 Gg yr$^{-1}$ by an inversion model (Simmonds et al., 2018a). Clearly, HFC-23 emissions have been increasing as a subsequence.

Currently, hydrofluorocarbons (HFCs) are widely used as substitutes of CFCs and HCFCs, because they have essentially no ODP and are therefore not contributing to ozone depletion. However, HFC-23 is not used as a substitute for CFCs or HCFCs,

but is used in halon-1301 (CBrF$_3$) production, semiconductor manufacturing, very low temperature refrigeration, and specialty fire extinguisher (Oram et al., 1998; Miller et al., 2010, Simmonds et al., 2018a), which means that emissions from deliberate use of HFC-23 are small. Hence, HFC-23 has mainly been vented from HCFC-22 production plants into the atmosphere (Montzka et al., 2019). Simmonds et al. (2018a) reported that global annual emissions of HFC-23 were estimated to have reached 13.3 ± 0.8 Gg yr$^{-1}$ in 2006 up from 4.2 ± 0.7 Gg yr$^{-1}$ in 1980 due to rising the production of HCFC-22. After 2006,

HFC-23 emissions rapidly decreased to 9.6 ± 0.6 Gg yr$^{-1}$ in 2009 as a result of thermal destruction of HFC-23 incentivized by the Clean Development Mechanism (CDM) under the Kyoto Protocol to the United Nations Framework Convention on Climate Change (UNFCCC). Due to a scheduled end of the CDM project, however, HFC-23 emission again increased, rapidly



reaching $14.5 \pm 0.6$ Gg yr$^{-1}$ in 2014 (Simmonds et al., 2018a). The annual global average mole fraction of HFC-23 reached 28.9 ppt (parts per trillion) in 2016, which corresponds to a radiative forcing of 5.2 mW m$^{-2}$. This is the second largest radiative forcing among all HFCs and fluorinated-gases just after HFC-134a (14.3 mW m$^{-2}$) (Montzka et al., 2019). Miller and Kuijpers (2011) suggested that if no additional abatement measures are implemented to reduce HFC-23, its emission will rise to 24 Gg yr$^{-1}$ in 2035, and the mole fraction will rise to 50 ppt which corresponds to a radiative forcing of 9 mW m$^{-2}$. Furthermore, if the emissions of HFC-23 are not regulated and all UNFCCC CDM project were terminated, the HFC-23 emission growth rate after 2030 would rise to 0.8 Gg yr$^{-2}$, which is four times larger than the previous trend (Miller and Kuijpers, 2011). In 2016, the parties to the Montreal Protocol agreed to amend the Montreal Protocol to gradually reduce the production and consumption of HFCs (the 2016 Kigali Amendment), and to control emissions of HFC-23. Unfortunately, Stanley et al. (2020) reported that the global HFC-23 emissions, derived from atmospheric measurements (top-down estimate), reached $15.9 \pm 0.9$ Gg yr$^{-1}$ in 2018 which was higher than in any year in history. Moreover, their results indicated that the top-down global emission in 2017 was $12.5 \pm 0.7$ Gg yr$^{-1}$ higher than the inventory-based emission of 2.4 Gg yr$^{-1}$ (bottom-up estimate), despite government mandated emission reductions in China and India. This result clearly implies that unreported HFC-23 by-product emissions exist. Therefore, the global observation system of atmospheric HCFC-22 and HFC-23 abundances is important to monitor the efficacy of the phase-down under the Montreal Protocol and to accurately project the impact of emissions of these compounds on ozone depletion and climate change into the future.

A ground-based in-situ measurement of HCFC-22 with a gas chromatography-mass spectrometer (GC-MS) technique was first reported by Rasmussen et al. (1980). Halocarbons and other Atmospheric Trace Species (HATS) group in Global Monitoring Division (GMD) of Earth System Research Laboratory in National Oceanic and Atmospheric Administration (NOAA/ESRL) has been analysing atmospheric minor constituents sampled in flasks at several remote sites since 1977, and the measurement of HCFC-22 by HATS group has started in 1992 (Montzka et al., 1993; Montzka et al., 2009). The Advanced Global Atmospheric Gases Experiment (AGAGE) observation network supported by a consortium of multinational institutions and organizations started HCFC-22 in-situ measurements in 1998 using a GC-MS (ADS systems since 1998 and more advanced "Medusa" systems since the mid-2000s) (Simmonds et al, 1995; Prinn et al., 2000; O'Doherty et al., 2004; Miller et al., 2008).

In contrast, the history of in-situ observation of atmospheric HFC-23 is relatively short. Atmospheric HFC-23 abundances were first reported in Oram et al. (1998) based on GC-MS measurements of flask background air samples collected at Cape Grim, Tasmania, Australia (40.7ºS, 144.7ºE), from 1978 to 1995. But high frequency in-situ measurement of HFC-23 by the AGAGE network are only available since the late 2000s using the GC-MS-Medusa at AGAGE stations (e.g. Cape Grim; Gosan, Jeju island, South Korea (33.3ºN, 126.2ºE)) (Miller et al., 2010; Kim et al., 2010; Simmonds et al., 2018a). Also, in-situ measurements of HFC-23 with AGAGE-compatible (but not identical) instruments have been operated at two stations of National Institute for Environmental Studies (NIES) in Japan: Hateruma, Okinawa (24.1°N, 123.8°E, since 2004) and Cape Ochiishi, Hokkaido (43.2ºN,145.5°E, since 2006) (Yokouchi et al, 2006; Fang et al., 2015). In total, there are, however, only 13 sites with HFC-23 in-situ measurements in the AGAGE network, including three affiliated stations.





Thanks to the evolution of molecular spectroscopy, and increasing atmospheric concentrations, space-borne remote sensing observation of HFCs, in addition to several CFCs and HCFCs, became possible (Nassar et al., 2006). For HFC-23, the first space-borne and balloon-borne remote sensing observations were done by the Atmospheric Chemistry Experiment-Fourier Transport Spectrometer (ACE-FTS) on SCISAT and the JPL MkIV interferometer, using the spectral region (1140–1160 cm⁻

$^1$) covering the $\nu_2$ and $\nu_5$ vibrational-rotational transition bands of HFC-23 (Harrison et al., 2012). Fernando et al. (2019) reported the HFC-23 trend above cloud-top derived from the ACE-FTS measurements for the period of 2004–2017, and indicated that the annual HFC-23 mole fractions retrieved from the ACE-FTS consistently averaged 5% smaller than ones at ground level from the AGAGE annual global mean dataset. However, the ACE-FTS observations do not have sensitivity to the troposphere where all HFC-23 emissions occur.

The Network for the Detection of Atmospheric Composition Change - Infrared Working Group (NDACC-IRWG) has been globally monitoring abundances of various atmospheric trace gases (e.g. $O_3$, HCl, $HNO_3$, $CH_4$, CO) using ground-based Fourier transform infrared spectrometer (FTIR) instruments (De Mazière et al., 2018). At present, the contributing ground-based FTIR instruments to the NDACC-IRWG are located at more than 20 sites around the world, and have yielded long-term consistent high-quality data by adherence to strict measurement and analysis procedures. For CFCs and HCFCs, for example, atmospheric

CFC-11, CFC-12, and HCFC-22 have been retrieved from infrared spectra taken by ground-based FTIRs at Reunion island (Zhou et al., 2016). For HFCs, however, there has been no attempt to retrieve their atmospheric abundances. If routine observations of atmospheric HFC-23 using the NDACC's ground-based FTIRs were possible, we could fill spatial and temporal gaps in the existing observations by AGAGE and ACE-FTS, which would allow for monitoring of global atmospheric HFC-23 abundances in greater detail than ever.

This study aims to investigate the retrieval procedure of atmospheric HFC-23 using the overlapping $\nu_2$ and $\nu_5$ vibrational-rotational transition bands of HFC-23. We analyze solar infrared spectra observed by two ground-based FTIRs installed at Rikubetsu, Japan, and Syowa Station, Antarctica. First, the details of the FTIR observations at both sites are described in Section 2. In Section 3, the retrieval strategy of HFC-23 for both sites is described in detail. Section 4 presents the results and characteristics of the HFC-23 retrievals, including the retrieval error budget. In Section 5, the time-series of our FTIR-retrieved

HFC-23 are compared to the in-situ measurements from the AGAGE network, and the modelled annual global mean dataset based on the AGAGE measurements. In addition, we discuss the HFC-23 trends derived from each dataset. Finally, conclusions and perspectives are summarized in Section 6.

## 2. FTIR observations

### 2.1 Rikubetsu

Measurements of atmospheric trace gases at Rikubetsu, Hokkaido, Japan (43.5ºN, 143.8ºE), have been carried out using two high spectral resolution FTIR instruments since May 1995, contributing to the NDACC-IRWG. The site is 200 km east of



Sapporo and located in a small town surrounded by forests and pastures. In October 1997, this observatory was relocated to the top of a hill (380 m a.s.l.) in the vicinity of the town. The first instrument, which operated until April 2010, was a Bruker IFS-120M FTIR spectrometer. In 2013, a Bruker IFS-120/5HR, an upgrade of the IFS-120HR, was installed as a second instrument taking over the observations by the IFS-120M, contributing to the Total Carbon Column Observing Network

(TCCON) in addition to the NDACC-IRWG. The FTIR instruments at Rikubetsu have taken solar infrared spectra from 500 to 7500 cm$^{-1}$ with a KBr beam splitter, the NDACC recommended optical filters, and two liquid nitrogen-cooled detectors: Indium-Antimonide (InSb) and Mercury-Cadmium-Telluride (HgCdTe, so-called MCT). Typically, measurements with 2 scans were executed to acquire a spectrum with resolution of 0.0035 cm$^{-1}$, however, sometimes 4 to 16 scans were co-added in order to improve signal-to-noise ratios (SNR). The spectra, covering long-term periods, have been used for various studies

of atmospheric tracers (e.g. $O_3$, CO, $C_2H_6$, and HCN) related to stratospheric composition change and biomass burning (Nakajima et al., 1997; Zhao et al., 1997; 2002; Koike et al., 2006; Nagahama and Suzuki, 2007).

In this study, we used the spectra measured with the NDACC optical filter #6 (covering 500–1400 cm$^{-1}$) and a MCT detector under clear-sky conditions since October 1997. For the observations with the NDACC filter #6 using the IFS-120/5HR from 2013 up to 2018, unfortunately, the SNR values of the spectra are about 20% of those achieved before the replacement of the

instrument because smaller apertures were used. Since 2019, the more suitable aperture size of 1.7 mm has been adopted for the measurements using the NDACC optical filter #6. Hence, those observed spectra were additionally used in the retrievals of HFC-23. Also, these spectra are degraded to 0.0070 cm$^{-1}$ (see Section 3.3).

### 2.2 Antarctic Syowa Station

Since the Japanese Antarctic Syowa Station (69.0°S, 39.6°E; 10 m a.s.l.) was established in 1957, various kinds of scientific observations (e.g. meteorology, upper atmospheric physics, cryospheric sciences, biology, geology) have continuously been conducted around the station. Syowa Station has been maintained by members of the Japanese Antarctic Research Expedition (JARE) each year. In 2007, a Bruker IFS-120M FTIR instrument was installed at Syowa Station by NIES and Tohoku University, in cooperation with the 48[th] JARE members. Measurements using the FTIR at Syowa Station contributed to

research related to stratospheric composition near the edge of the polar vortex during ozone hole evolution, due to its geographical location (Nakajima et al., 2020). As the station is a remote site in Antarctica, it is possible to observe the background atmosphere of the southern hemisphere which it is not influenced by local human activity.

The instrument had two liquid nitrogen-cooled detectors of InSb and MCT, which were the same as those in the FTIR at Rikubetsu. Solar infrared spectra (500–7500 cm$^{-1}$) were recorded using the same measurement settings as used at Rikubetsu,

under clear-sky conditions in 2007, 2011 and 2016, but not the polar night period. In this study, we used the spectra covering 500–1400 cm$^{-1}$ detected with the MCT detector. Note, that the observed spectra in 2007 were measured with the NDACC filter #6, but since 2011 the observations covering this spectral region were separated into two measurements using the narrower NDACC filter #7 (covering 500–1100 cm$^{-1}$) and #8 (covering 1000–1400 cm$^{-1}$). Similar to observations at Rikubetsu, these





measurements were used from 2 to 16 scans with 0.0035 cm$^{-1}$ resolution. However, we degrade the resolution of these spectra to 0.0070 cm$^{-1}$ (see Section 3.3).

## 3. Retrieval strategy of HFC-23

To derive HFC-23 vertical mole fraction profiles and total column abundances, all spectra taken from the FTIR instruments at Rikubetsu and at Syowa Station were analyzed with the SFIT4 version 0.9.4.4 program (see https://wiki.ucar.edu/display/sfit4/) based on the optimal estimation method (OEM) of Rodgers (Rodgers, 1976; Rodgers, 2000). This program was developed by scientists from the National Center for Atmospheric Research (NCAR), the University of Bremen, and other institutes taking part in the NDACC-IRWG, as an up-grade version of the previous SFIT2 algorithm

(Pougatchev et al., 1995). This program includes the procedure which calculates the theoretical absorption spectrum based on prior information (e.g. meteorological profiles, a priori profile of target) and fits the calculated spectrum to the observed one, for selected one or more spectral regions (micro-windows; MWs). Finally, the program derives the most suitable state vector (i.e. the retrieved target profile) that balances information obtained from observation and from the a priori. Hereafter, the details of HFC-23 retrieval are described.

### 3.1 Retrieval method

From the Rodgers's OEM, measured spectrum $y$ can be written using a forward model $\boldsymbol{F}$ with a vector vertical profile of gas $\boldsymbol{x}$ and all non-retrieved parameters (temperature, pressure, etc.) vector $\boldsymbol{b}$ as;

$$\boldsymbol{y} = \boldsymbol{F}(\boldsymbol{x}, \boldsymbol{b}) + \boldsymbol{\varepsilon}, \tag{1}$$

where $\boldsymbol{\varepsilon}$ is a measurement noise. By taking a Taylor's series expansion around a priori profile $\boldsymbol{x}_a$ and best estimated value $\widehat{\boldsymbol{b}}$ of $\boldsymbol{b}$, and neglecting higher orders, we get the linear expression of equation (1) as;

$$\boldsymbol{y} = \boldsymbol{F}(\boldsymbol{x}_a, \widehat{\boldsymbol{b}}) + \frac{\partial F}{\partial x}(\boldsymbol{x} - \boldsymbol{x}_a) + \frac{\partial F}{\partial b}(\boldsymbol{b} - \widehat{\boldsymbol{b}}) + \boldsymbol{\varepsilon} = \boldsymbol{y}_a + \boldsymbol{K}(\boldsymbol{x} - \boldsymbol{x}_a) + \boldsymbol{K}_b(\boldsymbol{b} - \widehat{\boldsymbol{b}}) + \boldsymbol{\varepsilon}, \tag{2}$$

where $\boldsymbol{y}_a$ is a spectrum calculated from a priori, $\boldsymbol{K}$ and $\boldsymbol{K}_b$ are weighting function matrices, which are often so-called Jacobian, for state vector $\boldsymbol{x}$ and model parameter $\boldsymbol{b}$, respectively. From the inversion of equation (2), we get the best estimated vertical

profile of gas mole fraction vector $\widehat{\boldsymbol{x}}$ as;

$$\widehat{\boldsymbol{x}} = \boldsymbol{x}_a + \boldsymbol{G}\boldsymbol{K}(\boldsymbol{x} - \boldsymbol{x}_a) + \boldsymbol{G}\boldsymbol{K}_b(\boldsymbol{b} - \widehat{\boldsymbol{b}}) + \boldsymbol{G}\boldsymbol{\varepsilon}, \tag{3}$$

where $\boldsymbol{G} = \partial \widehat{\boldsymbol{x}}/\partial \boldsymbol{y}$ is a gain matrix, whose line elements are so-called contribution function, which mean inversion sensitivity. Combining a profile $\boldsymbol{x}$ taken from an observed spectrum $\boldsymbol{y}$ with a priori profile as described in Rodgers (1976), if there is a linear relationship of $\boldsymbol{y} = \boldsymbol{K}\boldsymbol{x} + \boldsymbol{\varepsilon}$, the best estimation $\widehat{\boldsymbol{x}}$ is defined as following weighted average;

$$\widehat{\boldsymbol{x}} = (\boldsymbol{S}_a^{-1} + \boldsymbol{K}^T \boldsymbol{S}_\varepsilon^{-1} \boldsymbol{K})^{-1}(\boldsymbol{S}_a^{-1} \boldsymbol{x}_a + \boldsymbol{K}^T \boldsymbol{S}_\varepsilon^{-1} \boldsymbol{K}\boldsymbol{x}) = \boldsymbol{x}_a + \widehat{\boldsymbol{S}} \boldsymbol{K}^T \boldsymbol{S}_\varepsilon^{-1}(\boldsymbol{y} - \boldsymbol{K}\boldsymbol{x}_a), \tag{4}$$





where $S_a$ and $S_\varepsilon$ are a priori and measurement noise covariance matrices, respectively, $\widehat{S} = (S_a^{-1} + K^T S_\varepsilon^{-1} K)^{-1} = S_a K^T (S_\varepsilon + K S_a K^T)^{-1} S_\varepsilon K$ is a covariance matrix of $\widehat{x}$. Comparing the equation (3), which is neglected the error terms of the forward model parameters and the measurement noise, with equation (4), we get the following matrix, so-called averaging kernel matrix $A$;

$$A = GK = \frac{\partial \widehat{x}}{\partial x} = \widehat{S} K^T S_\varepsilon^{-1} K, \tag{5}$$

which is described in Rogers (2000) in detail. Each line in matrix $A$ is called the averaging kernel, which represents the sensitivity of retrieved value compared to the true value. The sum of diagonal elements of matrix $A$ (trace; tr($A$)) is called degrees of freedom for signal (DOFS), which gives the number of pieces of vertical information.

Since the forward model for FTIR observation is usually non-linear problem, $\widehat{x}$ is taken by minimizing the following cost function $J$ derived from Bayes' theorem and Gaussian statistics;

$$J(x) = (y - Kx)^T S_\varepsilon^{-1} (y - Kx) + (x - x_a)^T R(x - x_a), \tag{6}$$

where $R = S_a^{-1}$ is a regularization matrix. The second term of Equation (6) is generally called the constraint and it is important for solving stably the state vector $x$. In the case of Rodgers' OEM, the covariance matrix obtained from a realistic variability of target gas is used as the regularization matrix $R$, but we use Tikhonov regularization (Tikhonov, 1963) to set up $R$ in this study. The details about selection of the regularization matrix is described at Section 3.5. Finally, the cost function is minimized by the Gauss-Newton iteration method, so the appropriate profile is found by the iteration which is described as;

$$x_{i+1} = x_a + S_a K_i^T (S_\varepsilon + K_i S_a K_i^T)^{-1} [y - y_i + K_i(x_i - x_a)], \tag{7}$$

where $i = 0, 1, 2, \ldots,$ is the iteration counter, $K_i$ is the Jacobian diagnosed at $x_i$, and $y_i = F(x_i)$. If this iterative calculation converges, the best estimate of $\widehat{x}$ results.

### 3.2 Retrieval micro-windows

Table 1 summarizes the strategy for the retrieval of HFC-23 executed in this study. For the retrieval of HFC-23 from FTIR spectra, we used the $v_2$ and $v_5$ vibrational-rotational transition bands of HFC-23 located at ~1150 cm$^{-1}$, which is the same spectral region as the retrieval of ACE-FTS (Harrison et al., 2012). The infrared absorption by HFC-23 contributes typically to only about 1% of the atmospheric transmittance of solar infrared radiation at ground level (see Figure 3). Hence, choice of MWs is critically important for the retrieval from the ground-based measurement. To avoid three strong $H_2O$ absorption lines at 1149.47 cm$^{-1}$, 1151.54 cm$^{-1}$ and 1152.44 cm$^{-1}$, we used two MWs as; MW1: 1138.5–1148.0 cm$^{-1}$, MW2: 1154.0–1160.0 cm$^{-1}$. Major interfering species in these MWs are $O_3$, $N_2O$, $CH_4$, $H_2O$, HDO, $CCl_2F_2$ (CFC-12), $CHClF_2$ (HCFC-22), and $CH_3C(O)OONO_2$ (peroxyacetyl nitrate: PAN). Since there are several strong absorption lines of $O_3$ and $N_2O$ in these MWs, we retrieve profiles of these gases in addition to HFC-23. For the other species except for $CH_4$, we fit to an observed spectrum by scaling the a priori profile (column retrieval). In addition, $CH_3CCl_2F$ (HCFC-141b) and $CH_3CClF_2$ (HCFC-142b) exist as





minor interfering gases in the MWs, but these gases were not retrieved in this study because the contributions of these gases to the transmittance in the MWs are very small. More detail is described in the following.

### 3.3 Spectral correction and instrumental line shape

As was stated in Section 2, absorption spectra which include HFC-23 retrieval MWs were recorded with the NDACC #6 and #8 optical filters in the MCT channel with 0.0035 cm$^{-1}$ resolution. In order to reduce the spectral random noise, we degraded the spectral resolution from 0.0035 cm$^{-1}$ to 0.0070 cm$^{-1}$. Note that the zero-level of the measured spectra (see Figure 1) are raised (in maximum, about +5% relative to maximum signal intensity) and curved due to the non-linearity of the MCT detector. Therefore, we corrected this zero-level offset in measurement spectrum with second order polynomial fitting using

well-known absorption saturated bands sprinkled over the spectral region between 750–1350 cm$^{-1}$.

    On the other hand, the continuum level, which is equal to 100% in transmittance, was fitted by the following, because the shape of the continuum level is caused by the optical characterization of the FTIR instrument, especially the optical bandpass filter. Since the MWs for HFC-23 retrieval are rather wide, the slope and curvature (parabola) of the spectral continuum level over each MWs are retrieved in the SFIT4 program. This correction multiplies the transmission spectrum $\boldsymbol{B}$ by;

$$\boldsymbol{B} = \alpha(\boldsymbol{w} - w_0)^2 + \beta(\boldsymbol{w} - w_0) + 1, \tag{8}$$

where $\alpha$ is the curvature, $\beta$ is the slope factor, $\boldsymbol{w}$ is the wavenumber vector in the MW, and $w_0$ is the starting wavenumber of the MW. As a result, calculated spectrum $\boldsymbol{y}_c$ can be written as;

$$\boldsymbol{y}_c = \boldsymbol{B} \cdot \boldsymbol{\psi}[\tau(\boldsymbol{w})], \tag{9}$$

where $\tau(\boldsymbol{w})$ is a calculated transmission spectrum with absorptions by each gas and solar lines (the Fraunhofer lines), $\boldsymbol{\psi}[\tau(\boldsymbol{w})]$

is a transmission spectrum of $\tau(\boldsymbol{w})$ convolved with instrumental line shape (ILS) function.

    Hydrogen bromide (HBr) gas-cell spectra were taken using a mid-infrared internal light source in order to check the alignment of the FTIR instrument and to evaluate the ILS function for both the instruments at Rikubetsu and Syowa Station. At Rikubetsu, the first HBr cell spectrum was taken at 26 March 2002 after the relocation of the instrument in October 1997. In this study, all observed spectra from October 1997 to April 2010 with the IFS-120M instrument were convolved with the

ILS function derived from the HBr cell measurement, but for all observed spectra with the IFS-120/5HR instrument no ILS function was used because the instrument has always been maintained best optical alignment. At Syowa Station, HBr cell spectra were taken from time to time following installation and re-alignment. Therefore, ILS corrections were applied for all the spectra. The modulation efficiency and phase error of the ILS at Rikubetsu and Syowa Station were evaluated with LINEFIT9 and LINEFIT14 programs, respectively (Hase et al., 1999).





## 3.4 Spectroscopic parameters

For the calculation of absorption by each atmospheric chemical species, the HITRAN 2008 line-by-line spectroscopic database (Rothman et al., 2009) was primarily used. For spectroscopic parameters for $H_2O$ and its isotopes, the updated ATM18 line-list by one of us (G. C. Toon, NASA/JPL) was used (For detail, see https://mark4sun.jpl.nasa.gov/toon/atm18/atm18.html,

last access 8 August 2020). For heavy molecules (such as CFCs, HCFCs, HFCs, and PAN), there are no resolved line-lists available by HITRAN 2008.  For our retrieval of CFC-12, HCFC-22, HFC-23, and PAN, we used pseudo-line-list (PLL) developed by G. C. Toon (For detail, see https://mark4sun.jpl.nasa.gov/pseudo.html).  In these PLLs, the 296 K line strength and ground-state energy (*E''*) for each pseudo line were empirically reproduced by fitting transmittance laboratory spectra (absorption cross sections) acquired under various temperature and pressure conditions. In Harrison et al. (2012), the PLL of

HFC-23 obtained from the cross sections acquired with a resolution of 0.02 cm$^{-1}$ and a temperature range of 214–300 K and a total pressure range of 0.184–253 Torr by Chung (2005) were used to analyze solar occultation spectra, but there was a large systematic bias of ~30% in the retrieved profiles. This is dominantly caused by the poor quality of the used cross section dataset (e.g. inconsistency between the spectral absorptions and the temperature-pressure-mole fraction conditions). In order to reduce the systematic uncertainty in the HFC-23 PLL, Harrison (2013) reported new absorption cross section measurements

with a resolution of 0.015 cm$^{-1}$, which cover a wider spectral range of 950–1500 cm$^{-1}$ and more realistic atmospheric conditions in the troposphere and the stratosphere (i.e. a wider temperature range of 188–294 K and a wider pressure range of 23–762 Torr.)

For the current study, a new HFC-23 PLL was used with a wavenumber interval of 0.004 cm$^{-1}$ over a spectral range of 1105–1425 cm$^{-1}$ (https://mark4sun.jpl.nasa.gov/data/spec/Pseudo/CHF3_PLL_Update.pdf). In addition to the Chung's laboratory

spectra, this pseudo-line parameters were obtained from re-fitting Harrison's 2013 laboratory spectra, three spectra from the Pacific Northwest National Laboratory (PNNL) infrared database (Sharpe et al., 2004), and one spectrum from Gohar et al. (2004). Using this new PLL, which is dominated by the Harrison 2013 data, the bias in MkIV balloon measurements of HFC-23 is eliminated. In the forward model, the absorption line intensities are calculated by assuming the Boltzmann distribution which includes the temperature dependences of rotational/vibrational partition functions and induced emission. For the

rotational partition function, its temperature dependence is calculated from $(296/T)^\beta$, where $T$ is temperature and $\beta$ is temperature coefficient. For HFC-23, $\beta$ was set to 1.5, the normal value for non-linear molecules. To calculate the vibrational partition function, we assumed a harmonic oscillator approximation and used the fundamental vibrational frequencies and degeneracies from Ceausu-Velcescu et al. (2003). For solar lines, we used the empirical line-by-line parameters in mid-infrared region (Hase et al. 2006), which is in the SFIT4 program package.



### 3.5 Information of atmospheric state and regularization matrix

We consider 47 atmospheric layers for Rikubetsu, and 48 layers for Syowa Station from ground to 120 km in altitude. The thickness of the layers increases with height. We used Reanalysis-1 daily temperature and pressure data by National Center for Environmental Prediction (NCEP; http://www.ncep.noaa.gov) from the ground to 40 km, and zonal monthly-mean

climatological profiles by the COSPAR International Reference Atmosphere 1986 (CIRA-86) from 40 km to 120 km (Rees et al., 1990). For a priori profiles of atmospheric compositions, the averaged profiles in the period of 1980–2020 derived from the monthly-mean profile data computed by the Whole Atmospheric Community Climate Model (WACCM) version 6 (Chang et al., 2008) were basically used.

For HFC-23, the WACCM does not compute its profile and thus a priori profile of HFC-23 was based on globally and

annually mean mole fraction profile by two-dimensional chemistry-radiation-transport model by Naik et al. (2000). This a priori profile shows little decrease above the tropopause, reflecting a very long lifetime (228 years) of HFC-23 in the atmosphere (Montzka et al., 2019). For Rikubetsu, the HFC-23 a priori profile was scaled to 16 ppt at ground, which corresponds to the mole fraction of HFC-23 in 2002 in the northern hemisphere. For Syowa Station, the HFC-23 a priori profile was scaled to 24 ppt at ground, which corresponds to the mole fraction of HFC-23 in 2011 in the southern hemisphere.

For CFC-12, HCFC-22, HCFC-141b, and HCFC-142b, the mean profiles for 1995–2010 at Rikubetsu and for 2007–2016 at Syowa Station derived from the WACCM monthly dataset were used, because these species in the atmosphere have dramatically increased since 1980. Note that the mean profiles of HCFC-141b and HCFC-142b were used as fixed profiles in the HFC-23 retrieval.

For $H_2O$, HDO, and $CH_4$, a priori profiles were preliminarily retrieved (pre-retrieved) using other independent MWs,

because these profiles may induce large uncertainties in the HFC-23 retrieval. The detailed pre-retrieval procedure is described in Section 3.6.

In retrieval of atmospheric profile, it is crucial to select an optimal regularization matrix as a constraint on the a priori profile, because the regularization matrix affects the vertical resolution and the retrieval error. In the case of the general OEM, the regularization matrix $\boldsymbol{R}$ is the inverse of the a priori covariance matrix $\boldsymbol{S}_a$ which represents the natural variability for the target.

To calculate $\boldsymbol{S}_a$, the climatological dataset, which is constructed by a large number of independent profiles, should be used. For $O_3$, this is available because there are several high frequency observations (e.g. balloon-borne sondes, satellite measurements). In many cases, however, it is difficult to calculate a realistic natural variability for a priori profile, and our target gas also is one of them. Therefore, $\boldsymbol{S}_a$ is set up by an *ad hoc* method. In this study, Tikhonov regularization (Tikhonov, 1963) was used as in the previous studies of Sussman et al. (2009) for water vapor and Vigouroux et al. (2009) for formaldehyde

(HCHO), and the regularization matrix is defined as $\boldsymbol{R} = \alpha \boldsymbol{L}^T \boldsymbol{L}$, where $\alpha$ is the strength parameter of the constraint and $\boldsymbol{L}$ is a discrete derivative operator. We used the discrete first-order derivative operator $\boldsymbol{L}_1$ as $\boldsymbol{L}$:





$$L_1 = \begin{pmatrix} -1 & 1 & 0 & \cdots & 0 \\ 0 & -1 & 1 & \ddots & \vdots \\ \vdots & \ddots & \ddots & \ddots & 0 \\ 0 & \cdots & 0 & -1 & 1 \end{pmatrix}. \tag{10}$$

The operator respects the vertical shape of the a priori profile and suppresses oscillation of the retrieved profile. We have to properly determine the value of the regularization parameter $\alpha$, which is tuned to balance the constraint on the a priori profile and the residual between the measured and the calculated spectra, so-called the L-curve method (Hansen, 1992). In this study,

we were tuning $\alpha$ following the alternative method described in Section 4.D of Steck (2002). This method can determine the optimal $\alpha$ to minimize the total retrieval error (the smoothing plus the measurement errors; for details, see in Section 4.2). In this study, we used $\alpha = 100$ for all retrievals at Rikubetsu and Syowa Station.

According to Equation (6), we can understand that the measurement noise covariance matrix $S_\varepsilon$ is also a key constraint that balances the observations against the regularization matrix. The real SNR of the measured spectrum is an indicator of the noise

level for each spectrum, but for each retrieval the *ad hoc* SNR is used to determine $S_\varepsilon$. The *ad hoc* SNR is defined as the inverse of the root-mean-square (RMS) value of the residuals in a spectral fit (referred to as the fitted residuals). The *ad hoc* SNR is smaller than the real one since the fitted residuals are caused by various imperfections in forward model parameters (e.g. spectroscopic data, temperature profile, ILS) in addition to simple measurement noise. It is assumed that $S_\varepsilon$ is a diagonal matrix, and we put $SNR^{-2} = (y_m - y_c)^T (y_m - y_c)/N$ in the diagonal elements of $S_\varepsilon$, where $y_m$ and $y_c$ are the measured and

the calculated spectrum, respectively, and $N$ is the number of spectral points.

### 3.6 Pre-retrievals for H₂O, HDO, and CH₄

The vertical gradient and spatial-temporal variability of water vapor in the atmosphere are very large. In many cases with ground-based FTIR observations, it is impossible to choose the retrieval MWs without absorption structures of water vapor

and its isotopes, and thus it is important to use accurate water vapor profiles that are coincident with the location and time of each observation. Many previous studies (Vigouroux et al., 2009; 2012; Ortega et al., 2019) used the pre-retrieved $H_2O$ (and/or HDO) profiles using the dedicated MWs in order to reduce their interference errors. In this study, a priori profiles of $H_2O$ and HDO were acquired by pre-retrievals using the different MWs shown in Table 2.

$H_2O$ profiles were retrieved by using the MW of 824.40–825.90 cm$^{-1}$ which has been shown in the Atlas (Meier et al., 2004)

and the monthly profiles derived from the WACCM version 6 in the period of 1980–2020 as a priori profile for each spectrum. Since an $H_2O$ absorption line having $E''$ of 586.48 cm$^{-1}$ is in the MW, we assume that the uncertainties of the retrieved $H_2O$ profiles caused by temperature dependence of line strength are small. The $H_2O$ line is relatively weak and is hardly ever saturated, even when the humidity at Rikubetsu is high in summer. For HDO, the profile was pre-retrieved using the MW of 1208.40–1209.10 cm$^{-1}$, and the pre-retrieved $H_2O$ profile shape was used as a priori profile shapes of HDO and $H_2O$. This

HDO MW was used in the study of Vigouroux et al. (2012), but our MW is slightly wider than the previous study because the DOFS for HDO was increased when the wider window used. We estimated the retrieval uncertainties of approximately 10%





for the pre-retrieved total columns of $H_2O$ and HDO, which were mainly due to the systematic uncertainties of the spectroscopic parameters based on the HITRAN 2008.

It is important in deriving the HFC-23 retrieval that the interfering $CH_4$ profile is pre-retrieved with the dedicated MW shown in Table 2 because the weak $CH_4$ absorption structure in the retrieval MWs of HFC-23 could disturb estimation of the

true condition of HFC-23. Figure 2 shows the time-series of the total columns of HFC-23 and $CH_4$ retrieved from the FTIR infrared spectra observed at Syowa Station in 2007 and their scatter plot. The HFC-23 total column amounts (red-x plots) derived from retrievals of HFC-23 accompanied by column-retrieval (scaling) of $CH_4$ profile, and the scaled $CH_4$ total columns (green-x plots) are presented in Figure 2 (a). There is an anti-correlation between these two time-series. Since the typical seasonal cycle of $CH_4$ shows a minimum in summer due to destruction by the OH radical, the seasonal cycle of the scaled $CH_4$

total columns in the retrievals is inconsistent with the expected cycle. Furthermore, a seasonal cycle in the HFC-23 total columns is observed, but this is not expected since the atmospheric lifetime of HFC-23 is very long and thus its variability due to atmospheric loss is very small. Figure 2 (b) shows that the scatter plot of the total columns of HFC-23 and $CH_4$ in Figure 2 (a). Examination by a two-side hypothesis testing with Student's t-distribution under a null hypothesis in which there is no correlation between these total columns, the anti-correlation between HFC-23 and $CH_4$ is statistically significant with a

significance level of 5% ($p$-value < 0.05). Figure 2 (c) shows the time-series of the independent retrieved $CH_4$ total columns using a spectral region from 1201.820 to 1202.605 $cm^{-1}$ from Meier et al. (2004) (green dots), and of the HFC-23 total columns from retrievals using the independent retrieved $CH_4$ profiles ($CH_4$-fixed retrievals; red dots). In contrast to the scaled $CH_4$ in Figure 2 (a), the independently retrieved $CH_4$ shows the expected seasonal cycle. As the result, there is no un-realistic cycle in the HFC-23 total columns analyzed by the $CH_4$-fixed retrievals. As can be seen in the scatter plot of Figure 2 (d), there is no

correlation between HFC-23 and interfering $CH_4$. Therefore, we decided to pre-retrieve the $CH_4$ profile with the independent window before the retrieval of HFC-23. For pre-retrieving the profile of $CH_4$, we used the mean $CH_4$ profile (1980–2020) derived from the WACCM and the pre-retrieved $H_2O$ profiles mentioned above.

In conclusion, the pre-retrieved profiles of $H_2O$, HDO, and $CH_4$ were used as a priori profiles ($H_2O$ and HDO) and fixed profile ($CH_4$) in the subsequent retrieval of HFC-23.

## 4. Results of HFC-23 retrievals

Figure 3 shows an example of a spectral fitting result for the two MWs (MW1 and MW2) for HFC-23. This typical fitting was for a spectrum observed by the IFS-120M FTIR spectrometer at Syowa Station on 9 November 2011 at 13:47 UTC with a solar zenith angle (SZA) of 67.3º. In this case, the absorption contribution of HFC-23 is about 1% relative to the total

transmittance around 1156 $cm^{-1}$, corresponding to a total column of $3.85 \times 10^{14}$ molecules $cm^{-2}$. The typical root-mean-square (RMS) of the fitted residual (observed minus calculated spectrum) is 0.34%.





A summary of all the HFC-23 retrievals with SFIT4 at Rikubetsu and Syowa Station is shown in Table 3. The retrievals at Rikubetsu are summarized for the periods of 1997–2010 and 2019–2020 due to use of different instruments. The retrievals without negative values in profile were counted into the number of observations as the "valid" number, and those results were used to calculate each statistic. About 6% of observations at Rikubetsu in 1997–2010 were rejected. On the other hand, almost

all of observations at Rikubetsu in 2019–2020 and at Syowa Station were used. The mean RMS of the fitted residuals with the one standard deviation ($1\sigma$) at Rikubetsu is $0.35 \pm 0.14\%$ and $0.27 \pm 0.03\%$ for the 1997–2010 and 2019–2020, respectively. The mean RMS with $1\sigma$ at Syowa Station is $0.43 \pm 0.38\%$. The mean HFC-23 total column with $1\sigma$ standard deviation at Rikubetsu increased from $(3.23 \pm 1.10) \times 10^{14}$ molecules cm$^{-2}$ in the 1997–2010 period to $(5.59 \pm 0.43) \times 10^{14}$ molecules cm$^{-2}$ in the 2019–2020 period due to the increase of atmospheric HFC-23.

In the following sections, we describe the vertical information and the error estimation of our HFC-23 retrieval.

### 4.1 Vertical information

As mentioned in Section 3.1, the vertical information content of the FTIR retrievals is characterized by the averaging kernel matrix $A$, defined by equation (5). Figure 4 shows typical averaging kernels of the HFC-23 retrieval for the same spectrum

shown in Figure 3. Each curve coloured according to the right colour-bar in Figure 4 represents the row value of the averaging kernel matrix on the corresponding vertical layer. All the retrievals, including the typical case in Figure 4, are sensitive to troposphere and lower stratosphere, having a sensitivity peak in averaging kernel at ~4 km. The full widths at half maximum of the averaging kernels are ~20 km, and the mean DOFS for both all retrievals at Rikubetsu and Syowa Station is approximately 1.0. We conclude that only one piece of vertical information (the total column) can be extracted in this study.

### 4.2 Error analysis

The retrieval error can be considered as the difference between the retrieved and the true state vector. Subtract the true state vector $x$ from Eq. (3) including the systematic forward model error $\varepsilon_f$, the difference is defined as the following equation:

$$\hat{x} - x = (A - I)(x - x_a) + GK_b\varepsilon_b + G\varepsilon_f + G\varepsilon, \tag{11}$$

where $I$ is an identity matrix; $\varepsilon_b = b - \hat{b}$ (Rodgers, 1990; 2000). The retrieval error consists of four parts: the smoothing error $(A - I)(x - x_a)$, the non-retrieved forward model parameter error $GK_b\varepsilon_b$, the forward model error $G\varepsilon_f$ and the measurement noise $G\varepsilon$. The smoothing error is caused by the lack of vertical sensitivity combined with uncertainty in $x_a$ and includes the uncertainties from target gas, interfering gases and any other retrieved parameters (e.g. background correction parameters). The forward model parameter error $GK_b\varepsilon_b$ comes from the uncertainties of the parameters (e.g. profiles of temperature and

pressure, line lists of target and interfering gases, SZA, etc.) that are used for the forward model calculation. The forward model error is from the uncertainty in the forward model itself relative to true physics. In this study, the forward model error





was ignored, because the physical process (radiation transfer, infrared absorption, etc.) in the SFIT4 algorithm has been established well in previous studies.

The smoothing random error from target gas profile retrieval is described by the covariance matrix

$$S_{\text{s,Tar}} = (A_{\text{Tar}} - I)S_{\text{a,Tar}}(A_{\text{Tar}} - I)^T, \tag{12}$$

where $A_{\text{Tar}}$ is a part of the full averaging kernel matrix $A$ where the row and column elements run over all target components; $S_{\text{a,Tar}}$ is the a priori covariance matrix. In general, $S_{\text{a,Tar}}$ should represent the natural variability of the target gas, but we don't know well the natural variability of HFC-23 profile due to the lack of vertically measurement data. Therefore, a variability matrix derived from the AGAGE in-situ/sampling measurement dataset was used at each site as a substitute of $S_{\text{a,Tar}}$. For Rikubetsu, the variability of 25% against the a priori profile (square of $0.25x_{\text{a}}$) based on the background air

sampling data at Cape Grim in the period of 1995–2009 (Simmonds et al., 2018b) was adopted to the diagonal elements of the variability matrix. For Syowa Station, the variability of 10% against the a priori profile (square of $0.10x_{\text{a}}$) computed from the non-polluted data of the AGAGE in-situ measurements at Cape Grim (https://agage2.eas.gatech.edu/data_archive/agage/gc-ms-medusa/complete/tasmania/, last access 12 August 2020) was set to the diagonal elements of the matrix. Note that the systematic uncertainty for the smoothing error was not considered because we assumed that the shape of the HFC-23 a priori

profile does not have a large altitudinal gradient as mentioned in Section 3.1.5. The smoothing random errors for the retrieval uncertainties from all interfering species and some other retrieval parameters (background slope and curvature correction, wavenumber shift, solar line shift, solar line strength, and simple phase correction) can be written by:

$$\varepsilon_{\text{ret}} = A_{\text{Tar,Int}}(x_{\text{t}}^{\text{Int}} - x_{\text{a}}^{\text{Int}}) + A_{\text{Tar,Oth}}(x_{\text{t}}^{\text{Oth}} - x_{\text{a}}^{\text{Oth}}), \tag{13}$$

where $A_{\text{Tar,Int}}$ is a part of the full averaging kernel matrix $A$ where the row elements run over all target components and the

column elements run over all interfering species; $A_{\text{Tar,Oth}}$ is a part of the $A$ matrix where the row and column elements run over all target and other parameter components, respectively; $x_{\text{t}}^{\text{Int}}$ and $x_{\text{a}}^{\text{Int}}$, $x_{\text{t}}^{\text{Oth}}$ and $x_{\text{a}}^{\text{Oth}}$ are the true and a priori state vectors of interfering species and other parameters, respectively. To estimate the retrieval errors from the interfering gases, the variabilities around the a priori profiles for $H_2O$ (HDO) were set to 10% and the ones for other species were set to the values calculated from the used WACCM datasets.

In order to estimate the non-retrieved forward model parameter error, the following covariance matrix $S_{\text{f}}$ is calculated:

$$S_{\text{f}} = (GK_{\text{b}})S_{\text{b}}(GK_{\text{b}})^T, \tag{14}$$

where $S_{\text{b}}$ is the model parameter covariance matrix, which is from the uncertainties in the model parameters. For the random and systematic uncertainties of temperature at Rikubetsu and Syowa Station, the uncertainties on the NCEP temperature profiles were assumed. The uncertainty of temperature at Rikubetsu is about 2 K in the troposphere, 2–10 K between the

tropopause and 60 km, and 10 K above 60 km. The uncertainty of temperature at Syowa Station is about 2.5 K in the altitude range from the surface to 20 km, 2.5–10 K between 20 and 60 km, and 10 K above 60 km. The SZA random uncertainty was assumed an uncertainty of 0.15º, considering measurement time. For HFC-23, $N_2O$, $O_3$, $H_2O$, and HDO, the uncertainties of the spectroscopic parameters (i.e. line intensity, $S_v$; air-broadening coefficient, $\gamma_{\text{air}}$; coefficient of temperature dependence for


$\gamma_{\mathrm{air}}$, $n_{\mathrm{air}}$) were also estimated. For the uncertainties of $S_\nu$, $\gamma_{\mathrm{air}}$, and $n_{\mathrm{air}}$ of HFC-23, we set 10%, 15%, and 15%, respectively, based on the PLL database (see https://mark4sun.jpl.nasa.gov/data/spec/Pseudo/CHF3_PLL_Update.pdf). With regard to heavy molecules like HFC-23, ground state energy $E''$ values, which are relevant to the temperature dependency of $S_\nu$, is empirically given, and then their uncertainties are larger than light molecules (e.g. $H_2O$, $O_3$). In addition, the $E''$ uncertainty

more affects $S_\nu$ at a cold site like Syowa Station. We assumed the error of 50 cm$^{-1}$ for the $E''$ values of the HFC-23 PLL, and estimated the uncertainties of 10% and 15% at Rikubetsu and Syowa Station, respectively, as the effect of the $E''$ error to $S_\nu$. For $N_2O$, $O_3$, $H_2O$ and HDO, the spectroscopic uncertainties were derived from the HITRAN 2008 database. The uncertainties for $N_2O$ and $O_3$ were set with 5%, 10%, and 5% for $S_\nu$, $\gamma_{\mathrm{air}}$, and $n_{\mathrm{air}}$, respectively. For $H_2O$ and HDO, we gave the uncertainty of 10% to each parameter.

The measurement error was calculated by the error covariance matrix $\boldsymbol{S}_{\mathrm{n}}$ defined as:

$$\boldsymbol{S}_{\mathrm{n}} = \boldsymbol{G}\boldsymbol{S}_{\varepsilon}\boldsymbol{G}^T, \tag{15}$$

where $\boldsymbol{S}_{\varepsilon}$ is the measurement noise covariance matrix. We adopted the square inverse of the ad hoc SNR for the diagonal elements of $\boldsymbol{S}_{\varepsilon}$ as mentioned in Section 3.5.

Furthermore, we estimated the impact of the interfering $CH_4$ to the HFC-23 retrieval because the retrieved HFC-23 total

column is affected by the retrieval uncertainty of the pre-fitted $CH_4$ profile. The uncertainties of the pre-retrieved $CH_4$ total columns are dominantly caused by the systematic uncertainties of its spectroscopic parameters. Considering the spectroscopic parameter uncertainty provided by the HITRAN2008 database, the mean uncertainties of $S_\nu$, $\gamma_{\mathrm{air}}$, and $n_{\mathrm{air}}$ on the pre-retrieved $CH_4$ total columns were approximately 5%, 4%, and 1%, respectively, at both sites. Since the MW for $CH_4$ pre-retrieval is closed to the HFC-23 MWs, these spectroscopic uncertainties on $CH_4$ are partly cancelled between both MWs. Therefore, we

assumed that the uncertainties of $S_\nu$, $\gamma_{\mathrm{air}}$, and $n_{\mathrm{air}}$ for $CH_4$ are 3%, 3%, and 1%, respectively, in the HFC-23 MWs. The effects of the $CH_4$ systematic uncertainties to the retrieved HFC-23 total column were calculated from Equation (14) using these uncertainties. On the other hand, the effect of the $CH_4$ random uncertainty to the retrieved HFC-23 was derived from the $1\sigma$ variability on the pre-retrieved $CH_4$ total columns. The $1\sigma$ standard deviations at Rikubetsu and Syowa Station were 4% and 3%, respectively. To quantity this uncertainty, we tested the HFC-23 retrievals by making the pre-retrieved $CH_4$ profiles scaled

by ±4% and ±3% at Rikubetsu and Syowa Station, respectively. Then we calculated the percent difference between the HFC-23 total column retrieved with the scaled $CH_4$ profile ("Scaled $CH_4$") and the ones retrieved with the no-scaled $CH_4$ profile ("Normal"). The percent difference $D$ is defined as the following equation:

$$D\,[\%] = \frac{TC_{\mathrm{HFC\text{-}23,Scaled\,CH_4}} - TC_{\mathrm{HFC\text{-}23,Normal}}}{(TC_{\mathrm{HFC\text{-}23,Scaled\,CH_4}} + TC_{\mathrm{HFC\text{-}23,Normal}})/2} \times 100, \tag{16}$$

where $TC_{\mathrm{HFC\text{-}23,Scaled\,CH_4}}$ and $TC_{\mathrm{HFC\text{-}23,Normal}}$ are the HFC-23 total columns retrieved with the Scaled $CH_4$ profile and the

Normal $CH_4$ profile, respectively.

Table 4 lists the mean contributions to the relative total retrieval errors on the retrieved HFC-23 total columns at Rikubetsu for the 1997–2010 period and Syowa Station for the 2007–2016 period. Assuming that each error is independent, the total errors on retrieved total columns are simply calculated from the square root of the square sum of the error components.



At Rikubetsu, the random and systematic errors are 15% and 24%, respectively. The random error is dominated by the measurement error of 12%, and the uncertainty of 7.3% on the pre-retrieved $CH_4$ profile. The relative random uncertainty on the $CH_4$ pre-retrieved profile decreases from about 10% to about 5% during the period of 1998–2010. It indicates that the random error has been decreasing with the increasing trend of atmospheric HFC-23. The systematic error is characterized by the $E''$ uncertainty of HFC-23, the $S_\nu$ uncertainties of HFC-23 and $CH_4$, and the $\gamma_{air}$ uncertainties of $H_2O$ and HDO.

At Syowa Station, the random and systematic errors are 8.6% and 19%, respectively. The random error mostly comes from the measurement uncertainty of 6.8%, and the $CH_4$ pre-retrieved profile uncertainty of 4.4%. The $CH_4$ pre-retrieved profile uncertainty reduces from 5% to 3% during the 2007–2016 period, similar to the result at Rikubetsu. The systematic error is mainly caused by the $E''$ uncertainty of HFC-23 and the $S_\nu$ uncertainties of HFC-23 and $CH_4$. In contrast with the retrieval at Rikubetsu, the contributions of the line parameters of $H_2O$ and HDO are small.

As the result, the mean total error for all the retrieved HFC-23 total columns at Rikubetsu for the 1997–2010 period and Syowa Station are 28% and 21%, respectively. In our HFC-23 retrieval strategy, the retrieval error is dominated by the systematic uncertainty of the line parameters, especially the $S_\nu$ uncertainties of HFC-23 and $CH_4$. The contribution of the random error caused mainly by the measurement noise is relatively small.

Figure 5 (a) shows the time-series of the FTIR-retrieved HFC-23 total columns with the total random errors at Rikubetsu and Syowa Station. Note that two high total columns at Syowa Station in 2016 come from temporal contamination of HFC-23 refrigerant used for Cryogenic Frost-point Hygrometer (CFH) sonde observation (Vömel et al., 2007) which was executed at the same place and days. We can see the increasing trend of the retrieved HFC-23 total columns, even taking into account the random retrieval errors on the total columns.

### 4.3 Impact of background correction

Since the width of the MWs for our HFC-23 retrieval is 9.5 cm$^{-1}$ for MW1 and 6.0 cm$^{-1}$ for MW2, the shape of continuum levels (transmittance of 1.0) in the observed spectra, which result from the characteristic of the optical filters, should be properly corrected (a so-called background correction). In this study, we used a 2nd-order polynomial (slope + curvature) for fitting of the background continuum shape for a wide MW. If a simple linear slope is employed for the background spectra, the HFC-23 total column is systematically biased toward negative. The difference between using a linear slope and of a 2nd-order polynomial was calculated using the same formula as Equation (16). At Rikubetsu, the mean percent difference was about -33% throughout the analysis period. At Syowa Station, the mean percent difference was about -10%, smaller than at Rikubetsu. These relative biases lead underestimation to the trend on the retrieved HFC-23 abundances compared to that from AGAGE in-situ measurements. Therefore, it is very important that the curvature is considered to the background correction.





### 5. Comparison with surface in-situ data

#### 5.1 Datasets

##### 5.1.1 Ground-based FTIR data

In this study, the fitted RMS residuals for most retrievals were less than 0.5% (Table 3, Figure 5 (b), and Figure 6). Figure
5 (b) shows the time-series of the fitted RMS residuals at Rikubetsu and Syowa Station, along with the SZA. In general, the
RMS values rise with increasing SZA due to a decrease in the SNR as shown in Figure 6. However, with SZA lower than 50º,
there are some observations with fitted RMS values exceeding 0.5% at Rikubetsu before 1999. This is caused by relatively
poor optical alignment of the FTIR instrument until April 1999 when a Bruker technician re-aligned the instrument. In the
following analysis, we basically use the FTIR-retrieved HFC-23 data filtered with the threshold of the fitted RMS (< 0.5%) in
order to make the FTIR-retrieved data as uniform in quality as possible. However, this threshold rejects most retrievals at
Syowa Station in winter when ground-based FTIR observations at large SZA generally give large RMS for the spectral fit
residuals due to weak solar intensity. Thus, for the retrievals at Syowa Station, we applied two fitted RMS thresholds depending
on the value of SZA: the thresholds are < 0.5% for SZA < 85º and < 1.5% for SZA of 85º or greater. Note that the high HFC-
23 abundances caused by the water vapor profile observations with a CFH-sonde at Syowa Station in 2016, as mentioned in
Section 4.2, are also excluded.

Since the HFC-23 retrievals have only one piece of vertical information, as already mentioned in Section 4.1, we consider
the dry-air column-averaged mole fractions $X_{\text{HFC-23}}$ as the following formula:

$$X_{HFC-23} = \frac{TC_{\text{HFC}-23}}{TC_{\text{dry}}} = \frac{TC_{\text{HFC}-23}}{\frac{P_{\text{s}}N_{\text{A}}}{gm_{\text{dry}}}-TC_{\text{H}_2\text{O}}\frac{m_{\text{H}_2\text{O}}}{m_{\text{dry}}}}, \tag{17}$$

where $TC_{\text{HFC}-23}$, $TC_{\text{dry}}$, and $TC_{\text{H}_2\text{O}}$ are the FTIR-retrieved HFC-23 total column, the dry-air total column, and the a priori
(pre-retrieved) $H_2O$ total column, respectively; $P_{\text{s}}$ is the surface pressure calculated from the NCEP reanalysis; $N_{\text{A}}$ is
Avogadro's constant; $g$ is the column-averaged acceleration; $m_{\text{dry}}$ and $m_{\text{H}_2\text{O}}$ are the mean molecular masses of dry-air and
$H_2O$, respectively. Finally, we calculated the monthly mean column-averaged $X_{\text{HFC-23}}$ at both sites.

##### 5.1.2 AGAGE in-situ and air archive measurements

The AGAGE instruments are based on gas chromatography coupled with mass spectrometry (GC-MS) and cryogenic sample
pre-concentration system, so-called "Medusa" systems. The GC-MS-Medusa systems, with 2-hourly sampling and cooling to
~-180 ºC, are operated at each AGAGE station (Miller et al., 2008; Arnold et al., 2012). For HFC-23, reported in-situ
measurements started in 2007, after HFC-23 contamination from the air pump module had been resolved by changing from
Viton to Neoprene diaphragms (KNF Neuberger UN05 pumps). The HFC-23 abundances at all AGAGE stations are reported
relative to Scripps Institution of Oceanography (SIO), SIO-07 primary calibration scales, in dry-air mole fractions. The absolute
accuracies of the HFC-23 measurements were liberally estimated to be -3 to 2% (Simmonds et al., 2018a).



For the comparison with the FTIR measurements at Rikubetsu and Syowa Station, we used the AGAGE in-situ measurement HFC-23 data at Trinidad Head, California, USA (THD, 41.1ºN, 124.2ºW) and Cape Grim, Tasmania, Australia (CGO, 40.7ºS, 144.7ºE), respectively. We downloaded the high frequency HFC-23 in-situ measurement dataset for THD and CGO and used the embedded pollution flags (P) to remove polluted data (https://agage2.eas.gatech.edu/data_archive/agage/gc-ms-

medusa/complete/, last access 24 August 2020) and then calculated daily median mole fractions. Note that there are no in-situ measurements at THD and CGO or other AGAGE sites before 2007 due to the HFC-23 pump contamination problems. Therefore, we additionally used annual global mean mole fractions of HFC-23 estimated by the AGAGE 12-box model, a 2-dimensional atmospheric chemistry and transport model (Simmonds et al., 2018a), where pre-2007 abundances are only based on HFC-23 dry-air mole fractions measured in the Cape Grim Air Archive (CGAA) samples (Simmonds et al., 2018b). These

data were taken from the Simmonds et al. (2018a; 2018b) papers.

### 5.2 Time-series and seasonal variation

Figure 7 shows the time-series of the monthly mean FTIR-retrieved $X_{HFC-23}$ at Rikubetsu and Syowa Station, along with the dry-air mole fractions from the AGAGE annual global mean dataset, the CGAA samples, and the in-situ measurements at

THD and CGO. The error bar on each monthly mean $X_{HFC-23}$ is a $1\sigma$ standard deviation around the monthly mean. The AGAGE annual global mean data and the CGAA data are plotted with the uncertainties reported by Simmonds et al. (2018a, 2018b). The FTIR-retrieved $X_{HFC-23}$ data at Rikubetsu during the whole period agree well with the AGAGE annual global mean and the CGAA data. However, the FTIR dataset at Rikubetsu has a peak during spring and summer of each year. On the other hand, the time-series at Syowa Station has a systematic underestimation of about 5 ppt (about 25% relative to the CGO in-situ data

in 2007) compared to the CGO in-situ data, and almost no significant seasonal cycle.

Figure 8 shows monthly mean de-trended $X_{HFC-23}$ values (in %, relative to the trend for all data) and $1\sigma$ standard deviations at Rikubetsu and Syowa Station. At Rikubetsu, the monthly mean de-trended $X_{HFC-23}$ rises rapidly from March to May, with a large fluctuation ($\pm$ 15–20%) within each month. On the other hand, the monthly mean $X_{HFC-23}$ values from December to February are mostly stable with a relatively small standard deviations of about ±10% and a value of 10–15% smaller than the

AGAGE in-situ measurements of HFC-23. As mentioned in Section 1, HFC-23 has a very long lifetime of 228 years and there is almost no sink for HFC-23 in the atmosphere, i.e. HFC-23 is chemically inactive in the atmosphere. In addition, the sources of HFC-23 exist in limited places on the ground. For example, there is no HCFC-22 production in Australia and therefore Cape Grim is not impacted by this major source of HFC-23. Consequently, we expect almost no seasonal variation of the HFC-23 dry-air mole fraction at any remote site as seen in the times-series of the THD and CGO measurements. At Syowa Station, the

seasonal cycle on the FTIR-retrieved $X_{HFC-23}$ is almost unrecognized in Figure 8 because the variability is smaller than the retrieval random error of about 10%. As a hypothesis, we suggest that the peaks at Rikubetsu during spring–summer were caused by atmospheric transport from somewhere emitting HFC-23. Several previous studies using FTIR observations of biomass burning-derived gases and a backward trajectory analysis method (Zhao et al., 1997; 2002; Nagahama and Suzuki,



2007), showed that the airmasses over northern Japan at 800–300 hPa level during April to November were mostly transported from the Eurasian continent. Furthermore, Koike et al. (2006) investigated the seasonal contribution from various sources of tropospheric carbon monoxide (CO) at Rikubetsu in 2001, using a 3-dimensional global chemistry transport model. Their study showed that, for CO levels at 1 km, the contribution from Asian fossil fuel combustion increases from early spring to summer

due to Asian pollutants transported by the weak southwesterly wind in summer. Figure 5 (a) of Simmonds et al. (2018a) illustrated that while for the developed countries (e.g. Europe, Japan, USA) the contribution to annual global emission of HFC-23 has been decreasing since 2000, annual Chinese emission has been rapidly increasing since late 1990s and the contribution of Chinese emission to global emission exceeded 50% in early 2000s. Considering this, we suggest that the peaks of $X_{\mathrm{HFC\text{-}23}}$ at Rikubetsu during spring–summer before and after about 2002 may come from the HFC-23 emissions in developed countries

and China, respectively. This postulated change in the location of Eurasian HFC-23 emissions needs to be examined with an inversion study, but this exceeds the focus of our study.

We also propose that the FTIR observations at Rikubetsu in December, January, and February (DJF) represent the baseline of the atmospheric HFC-23 at the site. Although the observations at Rikubetsu look consistent with the AGAGE measurements as seen in Figure 7, in fact, it is indicated in Figure 8 that the retrievals at Rikubetsu have a negative bias of about 15%. The

FTIR-retrieved $X_{\mathrm{HFC\text{-}23}}$ data at Rikubetsu in DJF are shown as green open circles in Figure 7. In Section 5.4, we derive the trends for the $X_{\mathrm{HFC\text{-}23}}$ data in DJF,  together with those for all $X_{\mathrm{HFC\text{-}23}}$ data. These trends are compared with the AGAGE measurements and whether the DJF dataset represents the background level of HFC-23 at Rikubetsu or not. For the trend analysis at Syowa Station, all $X_{\mathrm{HFC\text{-}23}}$ data are used due to no significant seasonal cycle. The negative biases occurred at both FTIR sites are described in the following section.

### 5.3 Negative bias on $X_{\mathrm{HFC\text{-}23}}$

The negative bias with respect to the AGAGE surface measurements of the FTIR-retrieved $X_{\mathrm{HFC\text{-}23}}$ at Syowa Station (about 25% in 2007) is larger than that at Rikubetsu (about 15% in 2007). The difference of 10% between both sites could be explained by (1) the latitudinal concentration difference and (2) the temperature dependency of the derived HFC-23 pseudo-line

parameter. With regard to the latitudinal concentration difference, Figure 1 of Simmonds et al. (2018a) shows that the difference between the in-situ measurements at mid-latitude in the northern and southern hemisphere is about 1 ppt (5% in 2007). For the remaining of 5% of our observed difference, by using the PLL we retrieved the HFC-23 mole fraction values from each spectrum of four laboratory measurement datasets which had been used to create the HFC-23 PLL (see Section 3.4), and then investigated the discrepancies between the retrieved mole fractions and the reported ones in the laboratory datasets.

We here represent the discrepancies by the HFC-23 mole fraction scaling factors (MFSFs). Figure 9 (a) shows the HFC-23 MFSFs at the spectral region from 1105–1240 cm$^{-1}$ plotted versus temperature. In an ideal spectroscopic parameter, the MFSFs in all temperature region would be 1. The red plots in regard to the laboratory spectral dataset of Harrison (2013) in Figure 9



(a) present the curved temperature dependency like a parabola taking a minimum value of ~0.95 at around 240 K. In other words, it means that a retrieved mole fraction from a spectrum measured at 240 K is about 5% smaller than the actual mole fraction. This is consistent with what the ACE-FTS HFC-23 time-series illustrated in Fernando et al. (2019), whose study used the Harrison's laboratory spectra, has a negative difference of average 5% in comparing with the annual global mean data

calculated by the AGAGE 12-box model. As the annual mean surface temperature at Syowa Station is about 260 K, it is assumed that this temperature dependency caused the negative bias of 5% at Syowa Station, in addition to the latitudinal concentration contrast. Also, this temperature dependency on the PLL probably has affected the HFC-23 total columns at Rikubetsu. As shown in Figure 9 (a), the MFSFs of the Harrison's laboratory measurements at > 260 K rapidly increase with raising temperature. Therefore, the temperature dependency may cause a part of the seasonal variation of HFC-23 at Rikubetsu

because the surface temperature at the site ranged from 260 K to 300 K approximately. For the unrealistic cycle of $X_{\mathrm{HFC\text{-}23}}$, with maximum in summer, caused by the PLL, we estimated an amplitude of about 1 ppt as peak-to-peak value.

Here we assess the large negative bias of 15% on the FTIR-retrieved $X_{\mathrm{HFC\text{-}23}}$ at both sites. As mentioned in Section 4.2, our HFC-23 retrieval was mainly affected by the spectroscopic parameter uncertainties of HFC-23, HDO, and $CH_4$. Therefore, it is reasonable to assume that the negative bias mainly comes from the systematic uncertainty on these spectroscopic parameters.

However, it is difficult to quantify the contributions of these parameters to the bias on $X_{\mathrm{HFC\text{-}23}}$. In order to resolve the negative bias, we suggest that new laboratory measurements are needed to improve the spectroscopic parameters of the HFC-23 PLL. The negative bias of 15% is consistent with the systematic uncertainty of the HFC-23 line intensity which is estimated by the error analysis in Section 4.2. We suggest that the systematic uncertainty is affected by the temperature and pressure conditions in measuring the laboratory spectra of HFC-23. Figure 9 (b) shows the conditions of the HFC-23 laboratory measurements of

Harrison et al. (2013). Harrison's laboratory measurements (total 27 measurements) cover the temperature and pressure region corresponding to the altitude from the surface to the stratosphere, but the number of the measurements corresponding to the lower troposphere (below 600 hPa level) is only 3. The typical surface temperatures at Rikubetsu and Syowa Station range from 260 K to 300 K and from 240 K to 280 K, respectively. Hence, at a pressure corresponding to the surface, there is no measurement in the temperature region covering the surface temperature at Rikubetsu, except for summer, and Syowa Station.

This lack of the measurements could result in a significant error in creating the HFC-23 pseudo-line parameters. Therefore, high-accuracy laboratory spectra of HFC-23 are required at various atmospheric conditions of lower troposphere in order to improve the pseudo-line parameters of HFC-23. In addition, this should be done to further understand the reason for the negative bias and the seasonal cycle by intercomparison with the retrievals using the observed spectra at other NDACC-IRWG ground-based FTIR sites around the world.

### 5.4 Trend analysis

Table 5 summarizes the HFC-23 annual changes, in ppt year$^{-1}$, computed from the monthly $X_{\mathrm{HFC\text{-}23}}$ at both FTIR sites and from the AGAGE datasets. The columns of Table 5 represent the trend in the time periods of 1997–2010, 1997–2020, 2007–





2016, and 2007–2020. The annual changes were calculated by linear regression, and any seasonal cycles were neglected. The uncertainty on each annual change represents the standard error of the slope estimated by linear regression. The trend regression line for the $X_{HFC-23}$ data at Rikubetsu in DJF over the 1997–2020 period shown in Figure 7 as green-dashed line is consistently lower than the AGAGE in-situ measurements at THD by about 3 ppt (about 15% relative to the AGAGE in-situ measurements

in 2007) as mentioned in Section 5.2. Although the annual change rate for all $X_{HFC-23}$ at Rikubetsu over the 1997–2010 period ($1.090 \pm 0.072$ ppt year$^{-1}$) is larger than the AGAGE annual global mean data ($0.820 \pm 0.011$ ppt year$^{-1}$) and the CGAA data ($0.805 \pm 0.006$ ppt year$^{-1}$) over the same period, the annual change rate calculated from only $X_{HFC-23}$ in DJF ($0.817 \pm 0.087$ ppt year$^{-1}$) is in good agreement with the AGAGE data and the CGAA data within the uncertainties. For the 1997–2020 period, the annual change rate at Rikubetsu in DJF is $0.806 \pm 0.044$ ppt year$^{-1}$, which is consistent with the one derived from the

AGAGE annual global mean dataset during 1997 to 2016 ($0.843 \pm 0.008$ ppt year$^{-1}$). For the 2007–2020 period, the annual change rate at Rikubetsu in DJF ($0.894 \pm 0.099$ ppt year$^{-1}$) is slightly smaller than that of the AGAGE in-situ measurements at THD over the 2007–2019 period ($0.984 \pm 0.002$ ppt year$^{-1}$). In contrast, the annual trends for all $X_{HFC-23}$ at Rikubetsu over the periods of 1997–2020 and 2007–2020 are $0.794 \pm 0.043$ and $0.528 \pm 0.086$ ppt year$^{-1}$, respectively, and these are smaller than those for other AGAGE datasets because there are many high $X_{HFC-23}$ at Rikubetsu before 2010. Considering with the above,

it is obviously indicated that the FTIR-retrieved $X_{HFC-23}$ data in DJF represents the baseline of the atmospheric HFC-23 at Rikubetsu.

At Syowa Station, the annual change rate over the 2007–2016 period ($0.823 \pm 0.075$ ppt year$^{-1}$) is consistent with the annual global mean dataset ($0.878 \pm 0.020$ ppt year$^{-1}$) and the AGAGE CGO in-situ measurements ($0.874 \pm 0.002$ ppt year$^{-1}$) over the same period.

Summarizing the above, the trends of the atmospheric HFC-23 retrieved with our strategy basically agree well with the trends derived from the AGAGE datasets used here, except for the absolute values of HFC-23. These results indicated that the ground-based FTIR measurement has a capacity to monitor the long-term trends of HFC-23.

### 6. Conclusions

We have developed a procedure for retrieving atmospheric column abundances of HFC-23 with ground-based FTIR, and the first HFC-23 retrievals were carried out using the infrared spectra taken from ground-based FTIRs at Rikubetsu (1997–2020) and Antarctic Syowa Station (2007–2016) with the SFIT4 retrieval software. The two retrieval micro-windows (1138.5–1148.0 cm$^{-1}$ and 1154.0–1160.0 cm$^{-1}$), encompassing the $v_2$ and $v_5$ vibrational-rotational bands of HFC-23, were selected to avoid strong $H_2O$ absorption lines at 1149.47 cm$^{-1}$, 1151.54 cm$^{-1}$ and 1152.44 cm$^{-1}$. The significant interfering species in the

micro-windows are $O_3$, $N_2O$, $CH_4$, $H_2O$, HDO, CFC-12, HCFC-22 and PAN. In particular, $H_2O$, HDO and $CH_4$ affect the HFC-23 retrievals. Due to large daily variabilities of $H_2O$ and HDO in the atmosphere, those a priori profiles were pre-retrieved with the individual dedicated MWs ($H_2O$: 824.40–825.90 cm$^{-1}$, HDO: 1208.40–1209.10 cm$^{-1}$) for each observed spectrum and





were then simply scaled in the subsequent HFC-23 retrievals. For a priori profiles of $CH_4$, in order to reduce the retrieval error resulting from competition between several weak absorptions of $CH_4$ and the weak HFC-23 absorption, a pre-retrieval with the dedicated MW of 1201.820–1202.605 cm$^{-1}$ was carried out for each spectrum and then these $CH_4$ profiles were fixed in the subsequent HFC-23 retrievals. Our HFC-23 retrieval was typically sensitive to the atmospheric layer from the surface to the lower stratosphere. However, its DOFS was only 1 and only total column amount can be retrieved. The mean HFC-23 total columns retrieved from the observed spectra at Rikubetsu over the periods of 1997–2010 and 2019–2020 were estimated as $(3.23 \pm 1.10) \times 10^{14}$ and $(5.64 \pm 0.59) \times 10^{14}$ molecules cm$^{-2}$, respectively. The mean HFC-23 total column at Syowa Station over the 2007–2016 period was $(3.69 \pm 1.35) \times 10^{14}$ molecules cm$^{-2}$.

We estimated the random/systematic retrieval errors for the FTIR-retrieved HFC-23 total columns assuming four error components – the smoothing error, other retrieved parameter error, non-retrieved model parameter error, and measurement noise error. The retrieval random/systematic errors at Rikubetsu and Syowa Station are 15%/24% and 8.6%/19%, respectively. The random errors at both sites mainly come from measurement noise and $CH_4$ pre-retrieved profile uncertainty. The systematic errors at both sites are dominated by the uncertainty of the spectroscopic parameters, in particular the spectroscopic uncertainties of HFC-23, $H_2O$, HDO, and $CH_4$. The total error for the retrieved HFC-23 total columns at Rikubetsu and Syowa Station are 28% and 21%, respectively.

The time-series of the FTIR-retrieved HFC-23 columns at Rikubetsu and Syowa Station show obviously increasing trends. The FTIR-retrieved $X_{\text{HFC-23}}$ at both sites were compared with four ground-based datasets of the CGAA samples, the AGAGE 12-box model, and the AGAGE in-situ measurements at THD and CGO. The trends of $X_{\text{HFC-23}}$ at Rikubetsu in DJF and Syowa Station are consistent with the trends derived from those datasets, but at Syowa Station there is a negative bias of 5 ppt compared to the AGAGE in-situ measurements at CGO. The time-series of the FTIR-retrieved $X_{\text{HFC-23}}$ at Rikubetsu has a seasonal cycle with a peak during spring to summer, but the $X_{\text{HFC-23}}$ at Syowa Station did not show a significant cycle. We suggest that the seasonal cycle of HFC-23 at Rikubetsu is caused by the transport of HFC-23 emitted from the Eurasia. We found that the minimum of the seasonal cycle occurred from December to February and represented the background concentration of HFC-23 at Rikubetsu at that time. The negative bias at Rikubetsu in DJF was 3 ppt compared to the AGAGE in-situ measurements at THD. We showed that the bias occurred at both sites and were caused mostly by the spectroscopic parameter uncertainties of HFC-23, $H_2O$, HDO, and $CH_4$. Therefore, these molecules mostly affect the HFC-23 retrieval. A solution for this bias problem may be found in new high-resolution laboratory spectra of HFC-23 measured under the atmospheric conditions of lower troposphere leading to an expected improvement of the HFC-23 spectroscopic parameters.

The annual change rate of the $X_{\text{HFC-23}}$ at Rikubetsu in DJF over the periods of 1997–2010 and 1997–2020 were $0.817 \pm 0.087$ and $0.806 \pm 0.044$ ppt year$^{-1}$, respectively, which are in good agreement with the trend derived from the annual global mean mole fractions by the AGAGE 12-box model over the same periods. The annual change rate at Syowa Station is $0.823 \pm 0.075$ ppt year$^{-1}$ over the 2007–2016 period, which is also consistent with the trend from the CGO in-situ measurements over the same period. The trend derived from the $X_{\text{HFC-23}}$ data retrieved with our retrieval strategy agreed with other ground-based measurements.



The present study demonstrates that ground-based FTIR measurements are capable of monitoring the long-term trend of HFC-23. If this FTIR measurement technique were extended to other NDACC ground-based FTIR sites around world, the measurements reported from these sites would complement the global AGAGE observations and may lead to insights about the atmospheric changes of HFC-23 and its role in global warming.

*Author contributions.* MT, HN, and IMu designed the retrieval strategy with the SFIT4. MT performed the retrievals and wrote the manuscript with comments and suggestions received from all co-authors. HN and MT performed FTIR observations at Syowa Station. TN was responsible for FTIR observations with IFS-120M instrument at Rikubetsu. IMo performed FTIR observations with IFS-120/5 HR instrument at Rikubetsu. GCT created the HFC-23 pseudo-line-list and provided advice on
the use of the line list. RFW and JM provided the AGAGE in-situ measurements at Trinidad Head. PBK and PJF provided the AGAGE in-situ measurements at Cape Grim. HJW is responsible for the AGAGE data archival. HN and IMu supervised this work.

*Acknowledgements.* We acknowledge Makoto Koike, Kazuo Hanano, and Nobuyuki Yokozeki for the operation of the FTIR
observations at Rikubetsu. We want to thank Kosuke Saeki and Takeshi Kinase for performing the FTIR observations at Syowa Station in 2007 and 2011, respectively. We are grateful to all the members of the 48[th] Japanese Antarctic Research Expedition (JARE-48), JARE-52, and JARE-57, and the National Institute for Polar Research (NIPR) for supporting for the FTIR observations at Syowa Station. This work was carried out by the joint research program of the Institute for Space-Earth Environmental Research, Nagoya University. FTIR operations of the Rikubetsu site are financially supported in part by the
GOSAT series project.

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





**Tables**

**Table 1. Summary of retrieval settings used for HFC-23 retrievals.**

| Micro-windows | MW1 | MW2 |
|---|---|---|
| Spectral region [cm$^{-1}$] | 1138.50–1148.00 | 1154.00–1160.00 |
| Profile retrieval | HFC-23, N$_2$O, O$_3$ | |
| Column retrieval | H$_2$O, HDO, CFC-12, PAN, HCFC-22 | H$_2$O, HDO, CFC-12, PAN |
| Pre-retrieval | H$_2$O, HDO, CH$_4$ | |
| Fixed species | CH$_4$, HCFC-141b, HCFC-142b | |
| Spectroscopic parameters | PLL (HFC-23, CFC-12, PAN, HCFC-22) | |
| | ATM18 (H$_2$O, HDO) | |
| | HITRAN2008 (others) | |
| Pressure and temperature | NCEP Reanalysis-1, CIRA86 | |
| A priori profiles (HFC-23) | Naik et al. (2000) but scaled to 16 ppt (Rikubetsu) / 24 ppt (Syowa Station) at surface | |
| A priori profiles (others) | Mean profiles in the period of 1995–2010 (Rikubetsu) / 2007–2016 (Syowa Station) from WACCM version 6 (CFC-12, HCFC-22, HCFC-141b, HCFC-142b) | |
| | WACCM version 6 mean profiles from 1980 to 2020 (except for the above) | |
| Signal-to-noise ratio (SNR) | Calculated from each observed spectrum | |
| Background correction | Slope, Curvature | |
| Instrumental line shape (ILS) | LINEFIT9/14 | |

**Table 2. Windows used for the pre-retrievals of H$_2$O, HDO, and CH$_4$. Profile-retrieved species are in bold characters.**

| Target species | Micro-windows [cm$^{-1}$] | Interfering species | References |
|---|---|---|---|
| **H$_2$O** | 824.40–825.90 | **O$_3$**, CO$_2$, C$_2$H$_6$ | Meier et al. (2004) |
| **HDO** | 1208.40–1209.10 | CH$_4$, N$_2$O, H$_2$O, CO$_2$, O$_3$, HNO$_3$, COF$_2$ | Vigouroux et al. (2012) |
| **CH$_4$** | 1201.820–1202.605 | **N$_2$O**, H$_2$O, O$_3$, HNO$_3$ | Meier et al. (2004) |



**Table 3: Statistic summary of the fitted SNRs, the root-mean-squares (RMSs) of the fitted residuals (observed minus calculated spectrum), the degree of freedom for signals (DOFSs) and the retrieved HFC-23 total columns at Rikubetsu and Syowa Station. The errors of the fitted RMSs, the DOFSs, and the total columns are the one standard deviation (1$\sigma$) around the averages. The numbers of the HFC-23 retrievals ($N$) are divided into two parts of a number of the retrievals used in this analysis (valid) and of total ones including rejected ones (total).**

| Site (instrument) | Period | $N$ (valid / total) | Mean fitted SNR (MW1 / MW2) | Mean fitted RMS [%] | Mean DOFS | Mean HFC-23 total column [$10^{14}$ molecules cm$^{-2}$] |
|---|---|---|---|---|---|---|
| Rikubetsu (IFS-120M) | 1997–2010 | 1081 / 1152 | 293 / 371 | 0.35 ± 0.14 | 1.0 ± 0.02 | 3.23 ± 1.10 |
| Rikubetsu (IFS-120/5HR) | 2019–2020 | 30 / 30 | 350 / 414 | 0.27 ± 0.03 | 1.0 ± 0.01 | 5.59 ± 0.43 |
| Syowa Station (IFS-120M) | 2007–2016 | 206 / 207 | 294 / 308 | 0.43 ± 0.38 | 1.0 ± 0.03 | 3.69 ± 1.35 |



**Table 4: Mean random and systematic uncertainties on FTIR-retrieved HFC-23 total columns at Rikubetsu and Syowa Station.**

| Site (period) | Rikubetsu (1997–2010) | | Syowa Station (2007–2016) | |
|---|---|---|---|---|
| Error component | Random error [%] | Systematic error [%] | Random error [%] | Systematic error [%] |
| Smoothing | 1.4 | | 0.56 | |
| Retrieved parameters | 0.15 | | 0.070 | |
| Interfering species | 2.8 | | 0.51 | |
| Measurement | 12 | | 6.8 | |
| Temperature | 3.8 | 3.8 | 1.2 | 1.2 |
| SZA | 1.1 | | 2.5 | |
| $S_\nu$ of HFC-23 | | 10 | | 10 |
| $E''$ of HFC-23 | | 10 | | 15 |
| $\gamma_{air}$ of HFC-23 | | 3.8 | | 3.7 |
| $n_{air}$ of HFC-23 | | 0.51 | | 0.59 |
| $S_\nu$ of $N_2O$ | | 0.16 | | 0.072 |
| $\gamma_{air}$ of $N_2O$ | | 4.4 | | 1.3 |
| $n_{air}$ of $N_2O$ | | 0.79 | | 0.30 |
| $S_\nu$ of $O_3$ | | 0.063 | | 0.037 |
| $\gamma_{air}$ of $O_3$ | | 0.13 | | 0.088 |
| $n_{air}$ of $O_3$ | | 0.054 | | 0.038 |
| $S_\nu$ of $H_2O$ | | 0.048 | | 0.055 |
| $\gamma_{air}$ of $H_2O$ | | 6.6 | | 2.1 |
| $n_{air}$ of $H_2O$ | | 0.24 | | 0.13 |
| $S_\nu$ of HDO | | 0.070 | | 0.069 |
| $\gamma_{air}$ of HDO | | 15 | | 2.3 |
| $n_{air}$ of HDO | | 0.47 | | 0.15 |
| $CH_4$ pre-retrieved profile | 7.3 | | 4.4 | |
| $S_\nu$ of $CH_4$ | | 5.8 | | 4.4 |
| $\gamma_{air}$ of $CH_4$ | | 0.038 | | 0.063 |
| $n_{air}$ of $CH_4$ | | 0.012 | | 0.026 |
| Subtotal | 15 | 24 | 8.6 | 19 |
| Total | 28 | | 21 | |



**Table 5:** HFC-23 annual changes and standard errors derived from monthly mean $X_{HFC-23}$ at Rikubetsu and Syowa Station, in ppt year$^{-1}$. The annual changes computed from the AGAGE annual global mean dataset, the CGAA air sample dataset, and the AGAGE in-situ measurements at THD and CGO are also listed for the same periods, unless indicated by other time frames lists in brackets.

| Annual change [ppt year$^{-1}$] | 1997–2010 | 1997–2020 | 2007–2016 | 2007–2020 |
|---|---|---|---|---|
| Rikubetsu (FTIR) | $1.090 \pm 0.072$ | $0.794 \pm 0.043$ | – | $0.528 \pm 0.086$ |
| Rikubetsu DJF (FTIR) | $0.817 \pm 0.087$ | $0.806 \pm 0.044$ | – | $0.894 \pm 0.099$ |
| Syowa Station (FTIR) | – | – | $0.823 \pm 0.075$ | – |
| Annual global mean (12-box model) | $0.820 \pm 0.011$ | $0.843 \pm 0.008$ (1997–2016) | $0.878 \pm 0.020$ (2007–2016) | |
| CGAA | $0.805 \pm 0.006$ (1997–2009) | – | – | – |
| THD (AGAGE in-situ) | – | – | $0.924 \pm 0.002$ | $0.984 \pm 0.002$ (2007–2019) |
| CGO (AGAGE in-situ) | – | – | $0.874 \pm 0.002$ | $0.928 \pm 0.001$ (2007–2019) |


# Figures

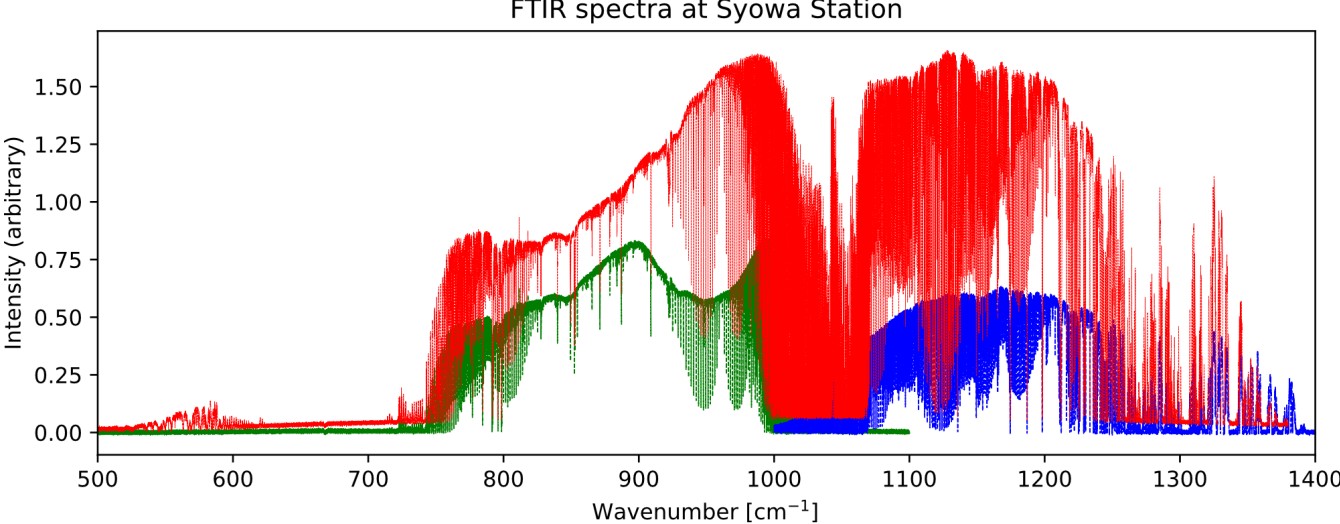

**Figure 1: Examples of solar absorption spectra taken from FTIR observations at Syowa Station. The red spectrum was obtained with the filter #6 on 30 September 2007. The green and the blue ones were measured with filter #7 and #8 on 30 September 2011, respectively. A positive zero-level offset is clearly seen on the red filter #6 spectrum.**







**Figure 2: Time-series of the total columns of HFC-23 and CH₄ retrieved from FTIR infrared spectra observed at Syowa Station in 2007. (a) HFC-23 total columns (red-x plots) derived from HFC-23 retrievals accompanied by column-retrieval (scaling) of CH₄ profile, and the scaled CH₄ total columns (green-x plots). (b) The correlation between HFC-23 and CH₄ of (a). (c) Independent retrieved CH₄ total columns using a spectral region from 1201.820 to 1202.605 cm⁻¹ (green dots), and HFC-23 total columns from retrievals using the independent retrieved CH₄ profiles as fixed profiles (red dots). (d) The correlation between HFC-23 and CH₄ of (c). Note that these retrieved HFC-23 columns were selected by the threshold of the fitted RMS value depending on the value of solar zenith angle (SZA): the threshold of the fitted RMS are < 0.5% for SZA < 85º and < 1.5% for SZA of 85% or greater.**



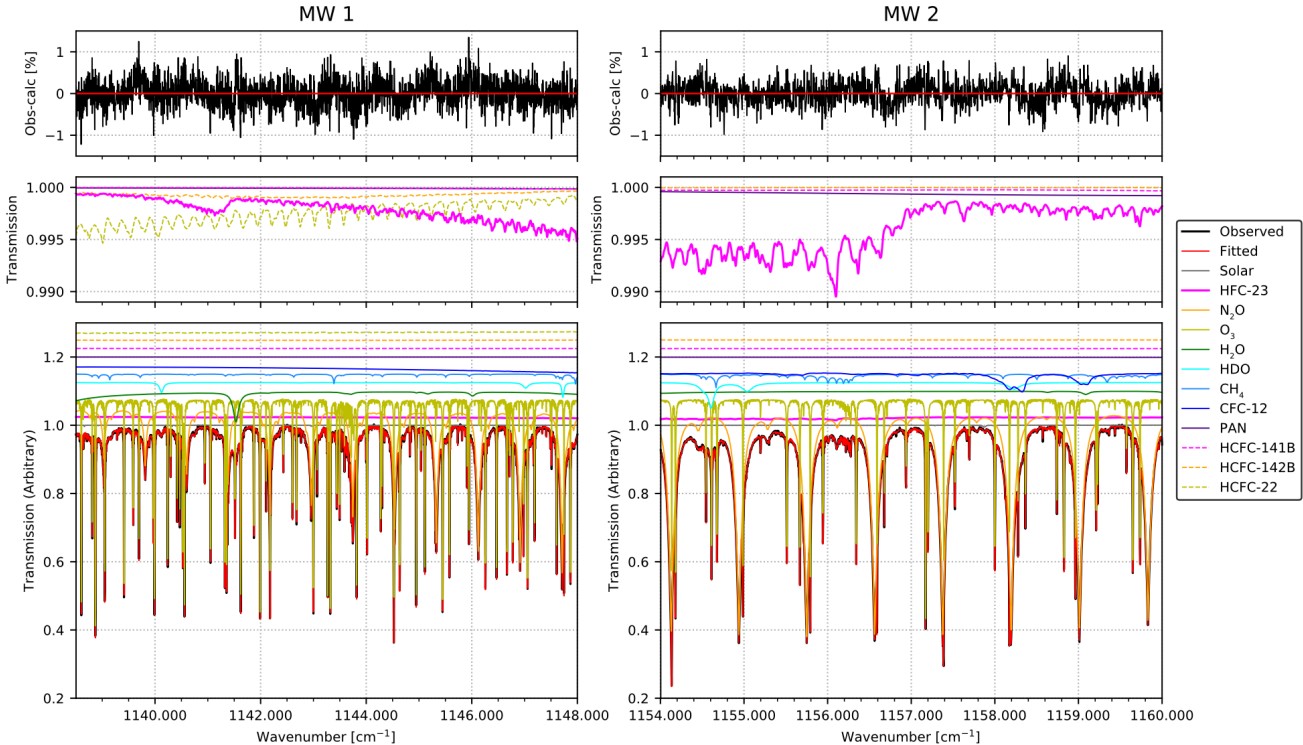

**Figure 3:** **Typical spectral simulation results of the two HFC-23 retrieval micro-windows (left panel: MW1; right panel: MW2) fitted to the observed spectrum at Syowa Station on 9 November 2011 at 13:47 UTC. The top two panels show the residuals (observed minus calculated) of the fittings for MW1 and MW2. The middle two panels show the absorption contributions of HFC-23, PAN, HCFC-141b, HCFC-142b, and HCFC-22 in MW1 and MW2. The bottom two panels show the individual contributions from each**
5 **interfering species, shifted by multiples of 0.025 for clarity, except the observed and the calculated lines.**

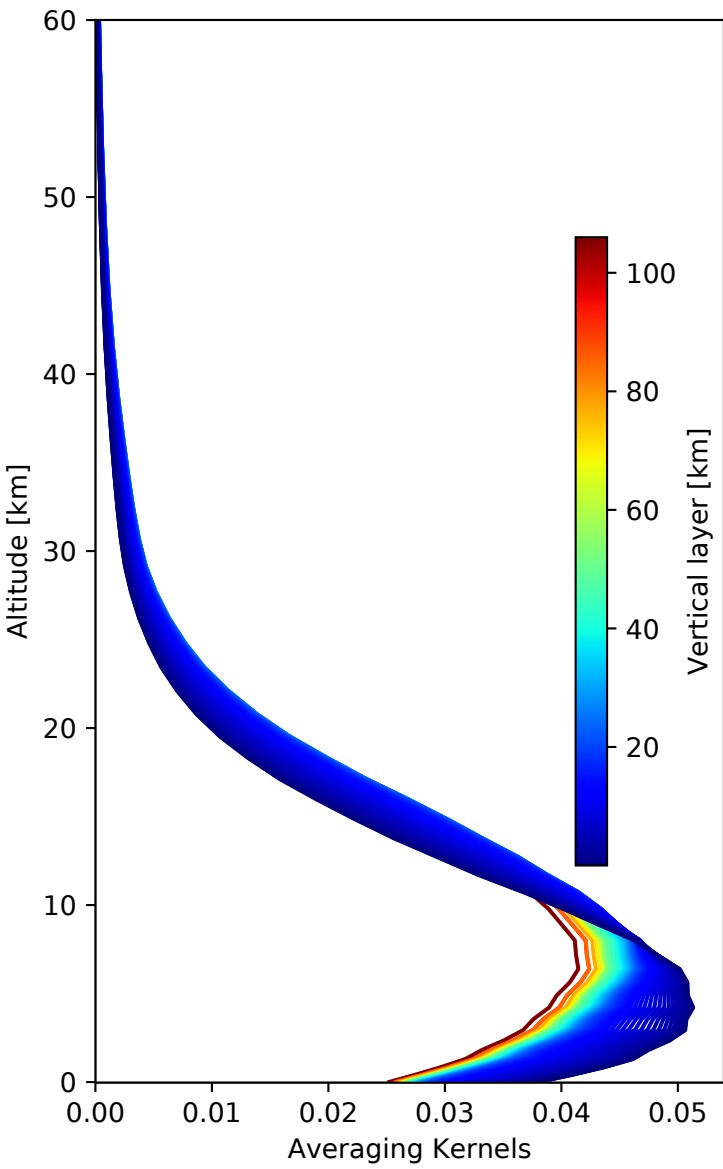

**Figure 4: Typical averaging kernels of the HFC-23 retrieval for the same spectrum shown in Figure 3, which are normalized using the a priori profile. Note that the vertical scale is from surface up to 60 km because there is almost no sensitivity above 60 km.**





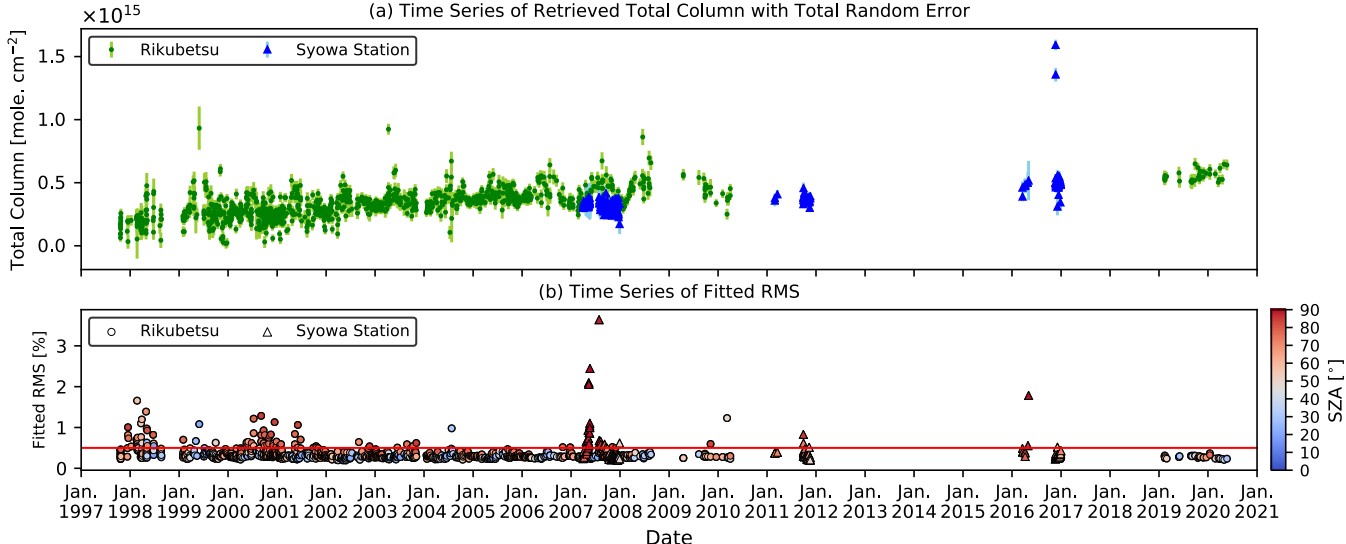

**Figure 5: (a): Time-series of FTIR-retrieved HFC-23 total columns with total random errors at Rikubetsu and Syowa Station. (b): The fitted RMS values on individual retrieved total column. The total columns at Rikubetsu and Syowa Station are shown by green circles and blue triangles, respectively. The fitted RMS values at Rikubetsu and Syowa Station are shown by circles and triangles, respectively, with the color-coding depended on the SZA.**

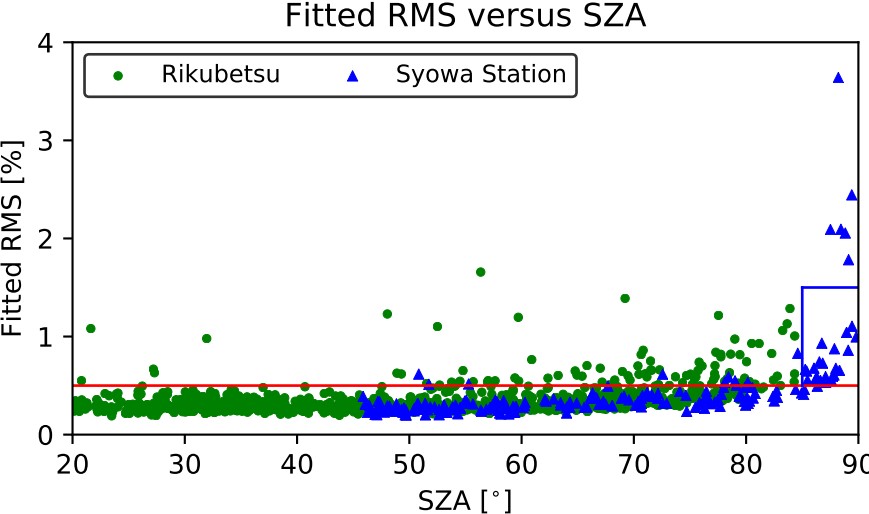

**Figure 6: The fitted RMS residuals versus the SZA values on individual retrieval. The RMS values at Rikubetsu and Syowa Station are shown by green circles and blue triangles, respectively.**





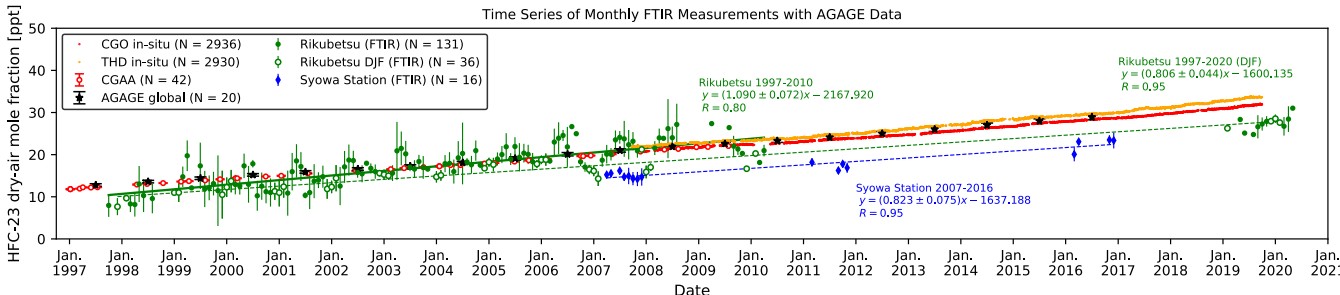

**Figure 7:** **Time-series of the monthly mean FTIR-retrieved $X_{\text{HFC-23}}$ at Rikubetsu and Syowa Station, along with the AGAGE in-situ measurements at CGO and THD, and the annual global mean mole fractions and the Cape Grim Air Archive samples, which were reported by Simmonds et al. (2018b).**

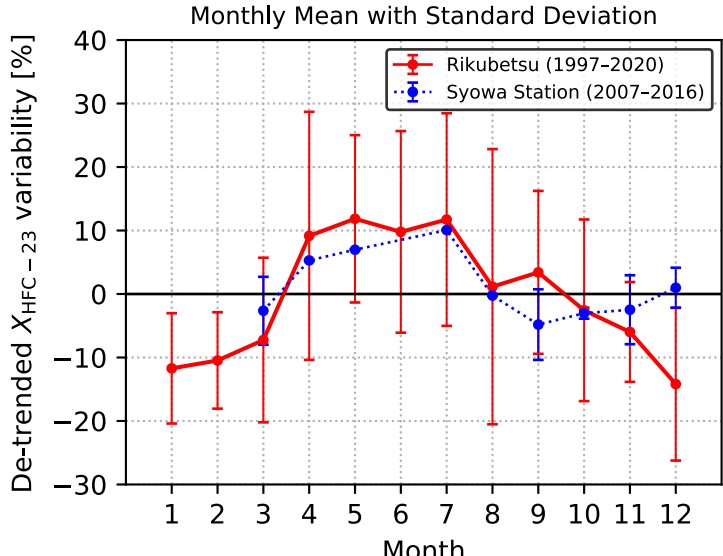

5    **Figure 8:** **Seasonal cycles of the FTIR-retrieved $X_{\text{HFC-23}}$ at Rikubetsu for the 1997–2020 period and at Syowa Station for the 2007–2016 period.**



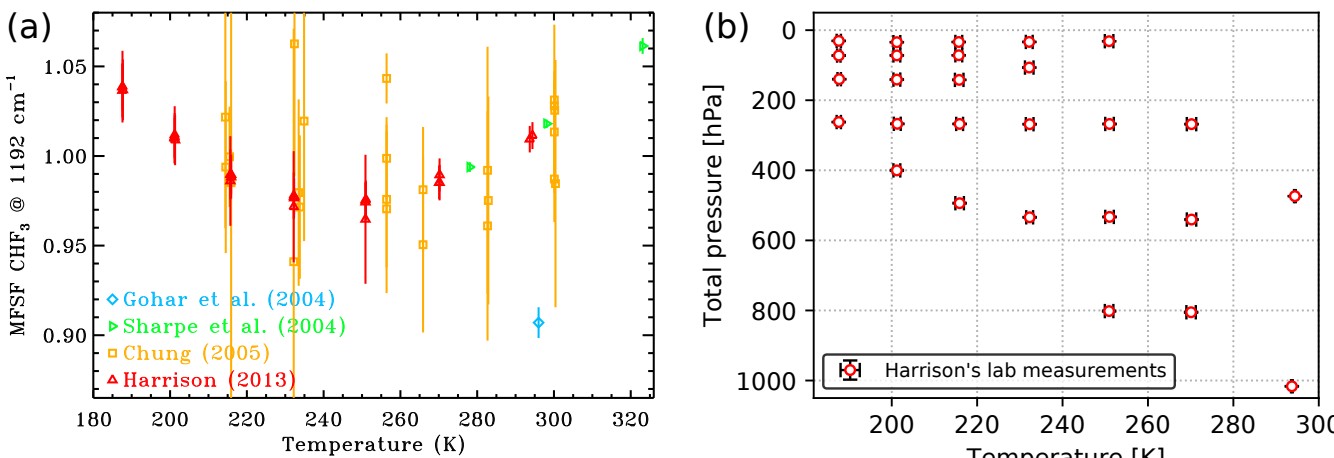

**Figure 9:** **(a): Retrieved mole fraction scaling factors from four HFC-23 laboratory spectrum datasets using the 2020 HFC-23 PLL at the spectral region from 1105–1240 cm⁻¹ plotted versus temperature. (b): The temperature and pressure conditions of the laboratory measurements of Harrison (2013).**