# Peer review of "First ground-based FTIR observations of HFC-23 at Rikubetsu, Japan, and Syowa Station, Antarctica"

_Atmospheric Measurement Techniques, 2020_

## Referee Comment (RC1)

The paper by Takeda et al reports the first attempt at a long term analysis of HFC-23 from groundbased FTIR. There are very compelling reasons to analyse this dataset based on the importance of HFC-23 as a strong greenhouse as (100 year GWP of over 12000), and the fact that it one of the HCFC family of chemicals whose production will not be phased out until 2030 under the Montreal Protocol for ozone protection. But its potential climate impact will remain for some time due to stockpiles and its long-lifetime. The addition of the NDACC network to long term measurements of this gas would be very welcome. However, this gas has not been a traditional target for this network as retrieving this gas from IR spectra is very difficult and challenging.

The purpose of this paper therefore is to outline, in some detail, how to retrieve total columns of HCFC-23 from IR spectra, obtain an annual trend and compare this trend against well-known and calibrated in-situ data. To retrieve this molecule is very difficult, so it will be of high interest to the NDACC FTIR community to follow this method. Indeed this will potentially unlock a worldwide dataset of measurements from Pole to Pole, linking in with the existing in-situ AGAGE network, and also offer a validation tool for satellite records.

The paper does indeed give a great deal of detail on the methodology, using one of the main software tools (SFIT4) used by the NDACC. This means the community can repeat this method without too many problems. However, the absorption signal from this species is so small (0.5%), and the spectra from filters 7/8 that are used throughout the network are relatively noisy, that without an independent dataset (AGAGE) the trends reported from this work would have been very difficult to justify.

The Japanese group leading this research have been involved in the NDACC for many years and therefore have significant experience in operating FTIR spectrometers and also analysing IR spectra.

The manuscript itself is a mixed bag in terms of its use of English language. The first 6 or so pages of text, abstract, sections 1 and 2, flow well and have been written well. However after this, section 3 has many instances of poorly written English, and use of English terms that are inappropriately used. Given that there are a number of English speakers in the coauthorship list, it would seem that they have not carefully commented and/or made changes in the manuscript that are necessary. This is an important paper to get out into the community so this referee has gone to the trouble of pointing out most of the issues. Some may have been missed; so coauthors read carefully!

This is an important paper for the FTIR community and a very useful outline of a method to analyse IR spectra for other researchers to follow. This paper therefore should be published, subject to the corrections/comments listed below

**Comments:**

Section 1 and 2: good background and well written. One change; section 2.2, line 29 had => has.

Section 3.1

- 1. Page 6, Line 20: "is a measurement noise." => "is the measurement noise."
- 2. Line 23: "which are often so-called Jacobian" => "also called the Jacobians"

- 3. Line 27: "is a gain matrix, whose line elements are so-called contribution function, which mean inversion sensitivity." => "is the gain matrix, or contribution function, which represents the sensitivity of the retrieved parameters to the measurement."
- 4. Line 29: "relationship of" => "relationship"
- 5. Page 7 line 2: "is a covariance" => "is the covariance"
- 6. Line 6-7: "Comparing the equation (3), which is neglected the error terms of the forward model parameters and the measurement noise" => "Comparing equation (3), which neglects the error terms of the forward model parameters and the measurement noise"
- 7. Line 9: "is usually non-linear problem," => "is usually a non-linear problem,"
- 8. Line 16: "profile is found by the iteration" => "profile is found by iteration"
- Page 7 line 2: "More detail is described in the following." => "More detail is described in the following sections."
- 10. Line 8: "in maximum, about +5% relative " => "about +5% relative"
- 11. Line 9: "offset in measurement spectrum with second order polynomial" => "offset in the measured spectrum with a second order polynomial"
- 12. Line 11-12: "On the other hand, the continuum level, which is equal to 100% in transmittance, was fitted by the following, because the shape of the continuum level is caused by the optical characterization of the FTIR instrument" => "On the other hand, the continuum level, which is equal to 100% in transmittance, has a shape that is caused by the optical characterization of the FTIR instrument"
- 13. Line 19: "(the Fraunhofer lines)," => "(the Fraunhofer lines), and"
- 14. Line 26: "has always been maintained best optical alignment." => "has always been maintained with the best optical alignment."
- 15. Page 9, line 26 : "is temperature coefficient" => "is the temperature coefficient"
- 16. Line 29: "in mid-infrared region" => "in the mid-infrared region"
- 17. Page 10, line 3: "pressure data by National" => "pressure data obtained from the National"
- 18. Line 5: "profiles by the COSPAR" => "profiles from the COSPAR"
- 19. Line 8: "were basically used." What does this mean exactly? Are there other profiles including HFC-23 that were not from WACCM?
- 20. Line 9-10: "For HFC-23, the WACCM does not compute its profile and thus a priori profile of HFC-23 was based on globally and annually mean mole fraction profile by two-dimensional"
  => "For HFC-23, WACCM does not provide a profile and thus the a priori profile of HFC-23 was based on the global and annual mean mole fraction profile by the two-dimensional"
- 21. Line 12: "ppt at ground," => "ppt at the ground,"
- 22. line 14: "ppt at ground" => "ppt at the ground"
- 23. Line 22: "In retrieval of atmospheric" => "In the retrieval of an atmospheric"
- 24. Page 11, line 2: "respects" => "conserves"
- 25. Lines 8-15. Please explain exactly how the Se "ad hoc" S/N was determined. In SFIT4 version 0.9.4.4 the actual S/N is recorded in the header of the input spectral file (commonly called the t15asc) for each micro-window. So while the Se matrix should in principle reflect the S/N for each spectral point, the implementation is such that a representative S/N for the whole window is applied. The ad hoc method is to scale this S/N to produce a retrieval that is both stable and has suitable dofs. This is part of the Steck procedure (4.D) to get  $\alpha$ . What was the effective S/N for an  $\alpha$  of 100?
- 26. Line 24: "which has been shown in the Atlas" => "as suggested in the NDACC IR reference microwindow Atlas"
- 27. Line 26: "Since an H2O absorption line having E'' of 586.48 cm-1 is in the MW" => "Since a H2O absorption line having an E'' of 586.48 cm-1 is in the MW"

- 28. Line 27: "temperature dependence of line strength are small" => "temperature dependence on the line strength are small"
- 29. Page 12, line 2: "on the HITRAN" => "on HITRAN"
- 30. Line 23: "and fixed profile" => "and a fixed profile"
- 31. Page 13, line 3: "values in profile" => "values in the profile"
- 32. Line 6: "with the one" => "within one"
- 33. Line 7: "with" => "within" ; both instances in this line.
- 34. Line 18: "for both all retrievals" => "for all retrievals"
- 35. Section 4.1: on the question of the dofs for HCF-23. The regularization strength  $\alpha$  is 100 which is quite strong. So this is a column scaling in effect, so the dofs will always end up being 1.0, regardless of how much information there is in the spectrum. There should be no surprise that the total column will be the only possible product from this method. That is, the dofs are being entirely driven by the method (Tikhonov), rather than the inherent spectral information.
- 36. Line 21: "Subtract" => "Subtracting"
- 37. Line 28: "from target gas" => from the target gas"
- 38. Page 14, line 2: "established well" => "well established"
- 39. Line 23-24: "and the ones for other species were set to the values calculated from the used WACCM datasets." => "and the uncertainties for other species were set to the values calculated from the appropriate WACCM datasets."
- 40. Line 28: "uncertainties on the" => "uncertainties reported from the"; NCEP has standard errors that are reported for their data which you can reference
- 41. Page 15 line 1: "we set" => "we set to"
- 42. Line 4: "and then their uncertainties" => "so that their uncertainties"
- 43. Line 5: "more affects" => "has a larger effect on"
- 44. Line 6: "error to" => "error on"
- 45. Line 12: as mentioned ealier above, what is the ad hoc S/N exactly quantitatively?
- 46. Line 19: "closed" => "close"
- 47. Lines 21&22: "to the" => "on the"
- 48. Line 31: the caption for table 4 is a little misleading. The word "uncertainty" is not correct, replace this with "error"
- 49. Page 16, line 17: "observation (Vömel et al., 2007) which was executed" => "observations (Vömel et al., 2007) which were flown"
- 50. Line 26: "and of a" => "and a"
- 51. Line 29: "biases lead underestimation to the trend on the retrieved" => "biases lead to an underestimation of the trend on the retrieved"
- 52. Line 30: "curvature is considered to the background correction." => "curvature of the continuum is considered when applying the background correction."
- 53. Page 17, line 17: "as the" => "as in the"
- 54. Page 18, line 6: : "to the HFC-23" => "to HFC-23"
- 55. Page 19, line 27 "remaining of" => "remaining"
- 56. Line 32: "region" => "regions"
- 57. Page 20, line 4: "negative difference of average" => "negative average difference of"
- 58. Line 10: "from 260 K to 300 K approximately." => "from approximately 260 K to 300 K."
- 59. Line 11: "ppt as" => "ppt as the"
- 60. Line 26: "of lower" => "of the lower"
- 61. Line 27: "this should be done to further understand the reason for the negative bias and the seasonal cycle by intercomparison with the retrievals using the observed spectra at other"

=> "further study should be undertaken to understand the reasons for the negative bias and apparent seasonal cycle by an intercomparison with HCF-23 total columns at other"

- 62. Page 21, line 3: "as green" => "as a green"
- 63. Lines 5, 7, 9,11: "annual change rate" => "annual growth rate"
- 64. Line 14: "Considering with the above, it is obviously indicated" => "Considering the above, it would seem"
- 65. Line 20: "strategy basically agree well" this is a very vague assertion. How do they agree? Perhaps it is better to write something like "strategy agree well within the errors"
- 66. Line 21: "These results indicated" => "These results indicate"
- 67. Page 22, line 22: "from the Eurasia." => "from Eurasia."
- 68. Line 28: "of lower" => "of the lower"
- 69. Lines 29,31: "annual change rate" => "annual growth rate"
- 70. Line 33: "ground-based measurements." => "ground-based in-situ measurements."
- 71. Page 31, table 4: It might be useful to add to this table the assumed uncertainties for each parameter that led to the computed error terms. Note the comment earlier about changing the caption. If however the uncertainties are included then the caption would read "errors and uncertainties"
- 72. Page 38, figure 7: what exactly are the R values computed with, ie, the FTIR data against all in situ data for each site?

---

## Author Comment (AC1)

**Reply to reviewer #1**

We thank anonymous reviewer #1 for his/her constructive review that would improve the contents of our paper. The review comments by anonymous reviewer #1 are numbered and repeated below as *in italic letters*, followed by our answers. In the new draft with corrections (supplement file), red, green, purple, and blue corrections are the revisions suggested by reviewers #1, #2, #3, and co-authors, respectively.

*<<Reviewer #1>>*

*<1-1> The paper by Takeda et al reports the first attempt at a long term analysis of HFC-23 from ground-based FTIR. There are very compelling reasons to analyse this dataset based on the importance of HFC-23 as a strong greenhouse as (100 year GWP of over 12000), and the fact that it one of the HCFC family of chemicals whose production will not be phased out until 2030 under the Montreal Protocol for ozone protection. But its potential climate impact will remain for some time due to stockpiles and its long-lifetime. The addition of the NDACC network to long term measurements of this gas would be very welcome. However, this gas has not been a traditional target for this network as retrieving this gas from IR spectra is very difficult and challenging.*

We agree.

*<1-2> The purpose of this paper therefore is to outline, in some detail, how to retrieve total columns of HCFC-23 from IR spectra, obtain an annual trend and compare this trend against well-known and calibrated in-situ data. To retrieve this molecule is very difficult, so it will be of high interest to the NDACC FTIR community to follow this method. Indeed this will potentially unlock a worldwide dataset of measurements from Pole to Pole, linking in with the existing in-situ AGAGE network, and also offer a validation tool for satellite records.*

We agree.

*<1-3> The paper does indeed give a great deal of detail on the methodology, using one of the main software tools (SFIT4) used by the NDACC. This means the community can repeat this method without too many problems. However, the absorption signal from this species is so small (0.5%), and the spectra from filters 7/8 that are used throughout the network are relatively noisy, that without an independent dataset (AGAGE) the trends reported from this work would have been*

*very difficult to justify.*

We agree.

*<1-4> The Japanese group leading this research have been involved in the NDACC for many years and therefore have significant experience in operating FTIR spectrometers and also analysing IR spectra.*

Thank you.

*<1-5> The manuscript itself is a mixed bag in terms of its use of English language. The first 6 or so pages of text, abstract, sections 1 and 2, flow well and have been written well. However after this, section 3 has many instances of poorly written English, and use of English terms that are inappropriately used. Given that there are a number of English speakers in the coauthorship list, it would seem that they have not carefully commented and/or made changes in the manuscript that are necessary. This is an important paper to get out into the community so this referee has gone to the trouble of pointing out most of the issues. Some may have been missed; so coauthors read carefully!*

We are sorry for the poor English after section 3. The English-native co-authors of this paper checked the draft carefully until section 2, but they didn't check the draft so carefully after section 3. In this course of revision, our English-native co-author checked the revised draft carefully throughout to the end, and it is now reflected in the revised draft as blue-colored changes. Anyway, we thank Reviewer #1 for his/her great effort to check the English of this draft especially after section 3.

*<1-6> This is an important paper for the FTIR community and a very useful outline of a method to analyse IR spectra for other researchers to follow. This paper therefore should be published, subject to the corrections/comments listed below*

Thank you. Our corrections for your individual comments are shown below.

*<1-7> Comments:*
*Section 1 and 2: good background and well written. One change; section 2.2, line 29 had => has.*

It was corrected as suggested.

*Section 3.1*

*1. Page 6, Line 20: "is a measurement noise." => "is the measurement noise."*

It was corrected as suggested.

*2. Line 23: "which are often so-called Jacobian" => "also called the Jacobians"*

It was corrected as suggested.

*3. Line 27: "is a gain matrix, whose line elements are so-called contribution function, which mean inversion sensitivity." => "is the gain matrix, or contribution function, which represents the sensitivity of the retrieved parameters to the measurement."*

It was corrected as suggested.

*4. Line 29: "relationship of" => "relationship"*

It was corrected as suggested.

*5. Page 7 line 2: "is a covariance" => "is the covariance"*

It was corrected as suggested.

*6. Line 6-7: "Comparing the equation (3), which is neglected the error terms of the forward model parameters and the measurement noise" => "Comparing equation (3), which neglects the error terms of the forward model parameters and the measurement noise"*

It was corrected as suggested.

*7. Line 9: "is usually non-linear problem," => "is usually a non-linear problem,"*

It was corrected as suggested.

*8. Line 16: "profile is found by the iteration" => "profile is found by iteration"*

It was corrected as suggested.

*9. Page 7 line 2: "More detail is described in the following." => "More detail is described in the following sections."*

It was corrected as suggested.

*10. Line 8: "in maximum, about +5% relative " => "about +5% relative"*

It was corrected as suggested.

*11. Line 9: "offset in measurement spectrum with second order polynomial" => "offset in the measured spectrum with a second order polynomial"*

It was corrected as suggested.

*12. Line 11-12: "On the other hand, the continuum level, which is equal to 100% in transmittance, was fitted by the following, because the shape of the continuum level is caused by the optical characterization of the FTIR instrument" => "On the other hand, the continuum level, which is equal to 100% in transmittance, has a shape that is caused by the optical characterization of the FTIR instrument"*

It was corrected as suggested.

*13. Line 19: "(the Fraunhofer lines)," => "(the Fraunhofer lines), and"*

It was corrected as suggested.

*14. Line 26: "has always been maintained best optical alignment." => "has always been maintained with the best optical alignment."*

It was corrected as suggested.

*15. Page 9, line 26 : "is temperature coefficient" => "is the temperature coefficient"*

It was corrected as suggested.

*16. Line 29: "in mid-infrared region" => "in the mid-infrared region"*

It was corrected as suggested.

*17. Page 10, line 3: "pressure data by National" => "pressure data obtained from the National"*

It was corrected as suggested.

*18. Line 5: "profiles by the COSPAR" => "profiles from the COSPAR"*

It was corrected as suggested.

*19. Line 8: "were basically used." What does this mean exactly? Are there other profiles including HFC-23 that were not from WACCM?*

Individual species which used WACCM 40-years averaged profile is now shown here ($N_2O$, $O_3$, and PAN) and the word "basically" was removed. The paragraph describing CFC-12 and HCFCs a priori profile is moved before the paragraph of HFC-23 a priori description.

*20. Line 9-10: "For HFC-23, the WACCM does not compute its profile and thus a priori profile of HFC-23 was based on globally and annually mean mole fraction profile by two-dimensional" => "For HFC-23, WACCM does not provide a profile and thus the a priori profile of HFC-23 was based on the global and annual mean mole fraction profile by the two-dimensional"*

It was corrected as suggested.

*21. Line 12: "ppt at ground," => "ppt at the ground,"*

It was corrected as suggested.

*22. Iine 14: "ppt at ground" => "ppt at the ground"*

It was corrected as suggested.

*23. Line 22: "In retrieval of atmospheric" => "In the retrieval of an atmospheric"*

It was corrected as suggested.

*24. Page 11, line 2: "respects" => "conserves"*

It was corrected as suggested.

*25. Lines 8-15. Please explain exactly how the Se "ad hoc" S/N was determined. In SFIT4 version 0.9.4.4 the actual S/N is recorded in the header of the input spectral file (commonly called the t15asc) for each micro-window. So while the Se matrix should in principle reflect the S/N for each spectral point, the implementation is such that a representative S/N for the whole window is applied. The ad hoc method is to scale this S/N to produce a retrieval that is both stable and has suitable dofs. This is part of the Steck procedure (4.D) to get α. What was the effective S/N for an α of 100?*

Sorry we don't have checked the SFIT4 core program in detail, but RMS value were used for Se in iteration process.    We corrected the explanation for Se.    S/N in t15asc is typically 2500 and effective S/N calculated from the RMS of the fit for α of 100 is typically 230-300.

*26. Line 24: "which has been shown in the Atlas" => "as suggested in the NDACC IR reference microwindow Atlas"*

It was corrected as suggested.

*27. Line 26: "Since an H2O absorption line having E" of 586.48 cm-1 is in the MW" => "Since a H2O absorption line having an E" of 586.48 cm-1 is in the MW"*

It was corrected as suggested.

*28. Line 27: "temperature dependence of line strength are small" => "temperature dependence on the line strength are small"*

It was corrected as suggested.

*29. Page 12, line 2: "on the HITRAN" => "on HITRAN"*

It was corrected as suggested.

*30. Line 23: "and fixed profile" => "and a fixed profile"*

It was corrected as suggested.

*31. Page 13, line 3: "values in profile" => "values in the profile"*

It was corrected as suggested.

*32. Line 6: "with the one" => "within one"*

It was corrected as suggested.

*33. Line 7: "with" => "within"; both instances in this line.*

It was corrected as suggested.

*34. Line 18: "for both all retrievals" => "for all retrievals"*

It was corrected as suggested.

*35. Section 4.1: on the question of the dofs for HCF-23. The regularization strength α is 100 which is quite strong. So this is a column scaling in effect, so the dofs will always end up being 1.0, regardless of how much information there is in the spectrum. There should be no surprise that the total column will be the only possible product from this method. That is, the dofs are being entirely driven by the method (Tikhonov), rather than the inherent spectral information.*

As discussed in Section 3.5, we used Tikhonov regularization rather than general optimal estimation method (OEM) in our sfit4 retrieval of HCF-23. Since we found that the DOFS was around 1.0 even if we apply OEM for the retrieval, we applied Tikhonov regularization method because it is more stable for the retrieval. The reason why we showed typical averaging kernel of HFC-23 retrieval is to show what altitudinal extent the HCF-23 retrieval has sensitivity.

*36. Line 21: "Subtract" => "Subtracting"*

It was corrected as suggested.

*37. Line 28: "from target gas" => from the target gas"*

It was corrected as suggested.

*38. Page 14, line 2: "established well" => "well established"*

It was corrected as suggested.

*39. Line 23-24: "and the ones for other species were set to the values calculated from the used WACCM datasets." => "and the uncertainties for other species were set to the values calculated from the appropriate WACCM datasets."*

It was corrected as suggested.

*40. Line 28: "uncertainties on the" => "uncertainties reported from the"; NCEP has standard errors that are reported for their data which you can reference*

It was corrected as suggested.

*41. Page 15 line 1: "we set" => "we set to"*

It was corrected as suggested.

*42. Line 4: "and then their uncertainties" => "so that their uncertainties"*

It was corrected as suggested.

*43. Line 5: "more affects" => "has a larger effect on"*

It was corrected as suggested.

*44. Line 6: "error to" => "error on"*

It was corrected as suggested.

*45. Line 12: as mentioned ealier above, what is the ad hoc S/N exactly quantitatively?*

It was corrected as suggested.

*46. Line 19: "closed" => "close"*

It was corrected as suggested.

*47. Lines 21&22: "to the" => "on the"*

It was corrected as suggested.

*48. Line 31: the caption for table 4 is a little misleading. The word "uncertainty" is not correct, replace this with "error"*

It was corrected as suggested.

*49. Page 16, line 17: "observation (Vömel et al., 2007) which was executed" => "observations (Vömel et al., 2007) which were flown"*

It was corrected as suggested.

*50. Line 26: "and of a" => "and a"*

It was corrected as suggested.

*51. Line 29: "biases lead underestimation to the trend on the retrieved" => "biases lead to an underestimation of the trend on the retrieved"*

It was corrected as suggested.

*52. Line 30: "curvature is considered to the background correction." => "curvature of the continuum is considered when applying the background correction."*

It was corrected as suggested.

*53. Page 17, line 17: "as the" => "as in the"*

It was corrected as suggested.

*54. Page 18, line 6: "to the HFC-23" => "to HFC-23"*

It was corrected as suggested.

*55. Page 19, line 27 "remaining of" => "remaining"*

It was corrected as suggested.

*56. Line 32: "region" => "regions"*

It was corrected as suggested.

*57. Page 20, line 4: "negative difference of average" => "negative average difference of"*

It was corrected as suggested.

*58. Line 10: "from 260 K to 300 K approximately." => "from approximately 260 K to 300 K."*

It was corrected as suggested.

*59. Line 11: "ppt as" => "ppt as the"*

It was corrected as suggested.

*60. Line 26: "of lower" => "of the lower"*

It was corrected as suggested.

*61. Line 27: "this should be done to further understand the reason for the negative bias and the seasonal cycle by intercomparison with the retrievals using the observed spectra at other" => "further study should be undertaken to understand the reasons for the negative bias and apparent*

*seasonal cycle by an intercomparison with HCF-23 total columns at other"*

It was corrected as suggested.

*62. Page 21, line 3: "as green" => "as a green"*

It was corrected as suggested.

*63. Lines 5, 7, 9,11: "annual change rate" => "annual growth rate"*

It was corrected as suggested.

*64. Line 14: "Considering with the above, it is obviously indicated" => "Considering the above, it would seem"*

It was corrected as suggested.

*65. Line 20: "strategy basically agree well" this is a very vague assertion. How do they agree? Perhaps it is better to write something like "strategy agree well within the errors"*

It was corrected as suggested.

*66. Line 21: "These results indicated" => "These results indicate"*

It was corrected as suggested.

*67. Page 22, line 22: "from the Eurasia." => "from Eurasia."*

It was corrected as suggested.

*68. Line 28: "of lower" => "of the lower"*

It was corrected as suggested.

*69. Lines 29,31: "annual change rate" => "annual growth rate"*

It was corrected as suggested.

*70. Line 33: "ground-based measurements." => "ground-based in-situ measurements."*

It was corrected as suggested.

*71. Page 31, table 4: It might be useful to add to this table the assumed uncertainties for each parameter that led to the computed error terms. Note the comment earlier about changing the caption. If however the uncertainties are included then the caption would read "errors and uncertainties"*

We agree to the reviewer. Table 4 and its caption is modified to show assumed uncertainty for each parameter and random/systematic errors which are computed by the assumed uncertainty.

*72. Page 38, figure 7: what exactly are the R values computed with, ie, the FTIR data against all in situ data for each site?*

No. The R values in Figure 7 are correlation coefficients of linear fit of each dataset.

[revised manuscript text omitted]

**Figures**

[Figure]

Figure 1: Examples of solar absorption spectra taken from FTIR observations at Syowa Station. The red spectrum was obtained with the filter #6 on 30 September 2007. The green and the blue ones were measured with filter #7 and #8 on 30 September 2011, respectively. A positive zero-level offset is clearly seen on the red filter #6 spectrum.

[Figure]

Figure 2: Time-series of the total columns of HFC-23 and CH$_4$ retrieved from FTIR infrared spectra observed at Syowa Station in 2007. (a) HFC-23 total columns (red-x plots) derived from HFC-23 retrievals accompanied by column-retrieval (scaling) of CH$_4$ profile, and the scaled CH$_4$ total columns (green-x plots). (b) The correlation between HFC-23 and CH$_4$ of (a). (c) Independent retrieved CH$_4$ total columns using a spectral region from 1201.820 to 1202.605 cm$^{-1}$ (green dots), and HFC-23 total columns from retrievals using the independent retrieved CH$_4$ profiles as fixed profiles (red dots). (d) The correlation between HFC-23 and CH$_4$ of (c). Note that these retrieved HFC-23 columns were selected by the threshold of the fitted RMS value depending on the value of solar zenith angle (SZA): the threshold of the fitted RMS are < 0.5% for SZA < 85º and < 1.5% for SZA of 85% or greater.

[Figure]

**Figure 3: Typical spectral simulation results of the two HFC-23 retrieval micro-windows (left panel: MW1; right panel: MW2) fitted to the observed spectrum at Syowa Station on 9 November 2011 at 13:47 UTC. The top two panels show the residuals (observed minus calculated) of the fittings for MW1 and MW2. The middle two panels show the absorption contributions of HFC-23, PAN, HCFC-141b, HCFC-142b, and HCFC-22 in MW1 and MW2. The bottom two panels show the individual contributions from each interfering species, shifted by multiples of 0.025 for clarity, except the observed and the calculated lines.**

[Figure]

**Figure 4:  Typical averaging kernels of the HFC-23 retrieval for the same spectrum shown in Figure 3, which are normalized using the a priori profile. Note that the vertical scale is from surface up to 60 km because there is almost no sensitivity above 60 km.**

[Figure]

**Figure 5:** **(a): Time-series of FTIR-retrieved HFC-23 total columns with total random errors at Rikubetsu and Syowa Station. (b): The fitted RMS values on individual retrieved total column. The total columns at Rikubetsu and Syowa Station are shown by green circles and blue triangles, respectively. The fitted RMS values at Rikubetsu and Syowa Station are shown by circles and triangles, respectively, with the color-coding depended on the SZA.**

[Figure]

**Figure 6:** **The fitted RMS residuals versus the SZA values on individual retrieval. The RMS values at Rikubetsu and Syowa Station are shown by green circles and blue triangles, respectively.**

[Figure]

[Figure]

**Figure 7: Time-series of the monthly mean FTIR-retrieved $X_{\text{HFC-23}}$ at Rikubetsu and Syowa Station, along with the AGAGE in-situ measurements at CGO and THD, and the annual global mean mole fractions and the Cape Grim Air Archive samples, which were reported by Simmonds et al. (2018b).**

[Figure]

**Figure 8:** Seasonal cycles of the FTIR-retrieved $X_{HFC-23}$ at Rikubetsu for the 1997–2020 period and at Syowa Station for the 2007–2016 period.

[Figure]

**Figure 9:** (a): Retrieved mole fraction scaling factors from four HFC-23 laboratory spectrum datasets using the 2020 HFC-23 PLL at the spectral region from 1105–1240 cm$^{-1}$ plotted versus temperature. (b): The temperature and pressure conditions of the laboratory measurements of Harrison (2013).

---

## Author Comment (AC2)

**Reply to reviewer #2**

We thank Dr. Chris Boone (reviewer #2) for his constructive review that would improve the contents of our paper. The review comments by reviewer #2 are numbered and repeated below as *in italic letters*, followed by our answers. In the new draft with corrections (supplement file), red, green, purple, and blue corrections are the revisions suggested by reviewers #1, #2, #3, and co-authors, respectively.

*<<Reviewer #2 (Chris Boone)>>*

*<2-1> Ground-based column HFC-23 observations are reported for two locations, retrieved from FTIR measurements using SFIT4. Error sources are evaluated, trends are determined and compared to results from the AGAGE network. The results from the Japanese station exhibit large seasonal fluctuations that are attributed to transport from Asia, but the Antarctic results show no significant seasonal variation. The wintertime (December-January-February) results from the Japanese station are 10-15% lower than the AGAGE 12-box model estimates, while the results from the Antarctic station are roughly 25% lower than AGAGE estimates. The discrepancies are attributed in part to deficiencies in the temperature dependence of the pseudolines employed in the calculation of HFC-23, coupled with the latitudinal differences for the Antarctic station. Trend values for the two stations show generally good agreement with results from equivalent AGAGE measurement sites, supporting the use of ground-based FTIR measurements for HFC-23 trend monitoring.*

*Overall, the results appear to be of good quality, although I am not sure I agree with the interpretation of the seasonal HFC-23 variation (more on that later). The article is well organized, and the writing is clear, albeit with a few instances of wording/grammar issues.*

Thank you.

*<2-2> I felt there was a missed opportunity here for placing the bias relative to the AGAGE results into a broader context. I agree with the authors' choice of using their winter (DJF) results as their "background level" to avoid the unexpected seasonal variation. On page 18, line 23, the authors state: values from December to February are mostly stable with a relatively small standard deviations of about +/-10% and a value of 10–15% smaller than the AGAGE in-situ measurements of HFC-23.*

*Looking at Figure 4 of the Fernando et al article referenced in this paper, the global average results from the ACE-FTS exhibit very similar behavior, a persistent low bias relative to AGAGE*

*results, roughly 10% in magnitude. Because the ACE-FTS employs the Harrison HFC-23 cross sections in its retrievals and the current study uses pseudo lines based primarily on the same cross sections, this agreement in the bias relative to AGAGE does not seem like a coincidence. It looks like there is a systematic discrepancy between results derived using the Harrison cross sections and AGAGE results. I believe absolute calibration of the Harrison cross sections made use of the PNNL measurements, and I would be surprised if that group's determination of HFC-23 concentrations were off by 10% or so. However, I have no knowledge of the measurement technique employed by AGAGE, so I have no idea if a ~10% error on that end is feasible. It seems clear HFC-23 derived using the Harrison cross sections are inconsistent with AGAGE measurements, but I have no insight into the source of that discrepancy.*

We agree to reviewer #2 on the systematic negative bias of HFC-23 derived using the Harrison cross sections (ACE-FTS) and pseudo lines (this study). Therefore, we added description of negative biases of -15 to -20% and -25%, and their possible cause due to the HFC-23 spectroscopy in the abstract.

*<2-3> I am somewhat skeptical that the seasonal variation in measured HFC-23 is real. No other station sees such behavior. They theorize that it comes from long range transport, and yet not a hint of it continues on across the Pacific Ocean to Trinidad Head, despite the apparent large magnitude of the emissions. It is possible the station in question is located in just the right location downwind of a large emitter, and perhaps the flow would routinely divert away from Trinidad head as it travels across the Pacific (not my area of expertise). Emissions during HCFC-22 production is a major source of HFC-23, and Japan is a significant HCFC-22 producer, so perhaps the emissions originate from near the station, which would explain why they see such large quantities. However, I would be inclined to think there might be an artifact in the retrieval. Are there systematic differences in the residuals for baseline HFC-23 conditions versus enhanced HFC-23 conditions? If additional structure appears in the residuals in the latter case, the retrieval may be compensating for something missing in the analysis or compensating for large residuals from some other constituent (e.g., from water lines if H2O levels are high). Is there a strong correlation of enhanced HFC-23 with H2O, temperature, the ratio of HDO toH2O, seasonal variations in CH4, or biomass burning products like HCN or C2H6? I suspect H2O might be the most dangerous of the bunch for impacting the retrieval. Correlation does not prove cause but may help identify the source of a retrieval artifact, if one exists.*

Japanese HFC-23 emission had already diminished at around 2004 as shown in Figure 2-1. We think there is almost no emissions originate from near the station in Japan after 2004.

[Figure]

Figure 2-1 Temporal variation of HFC-23 emission in Japan from HCFC-22

[https://www.env.go.jp/earth/ondanka/ghg-mrv/methodology/material/methodology_2B9_1.pdf]

And you can find spikey enhancement of HFC-23 especially in summer season in the surface measurement at Ochiishi, Japan in Figure 2-2. (Ochiishi is located in the same island as Rikubetsu. You can get the figure from https://gaw.kishou.go.jp/search/file/0053-2008-1502-01-01-9999).

[Figure]

Figure 2-2. Surface HFC-23 measurement in Ochiishi

[Figure]

Figure 2-3. Airmass trajectory to Rikubetsu in 2006.

We calculated 10-days backward trajectories originated from Rikubetsu at 2000 km altitude on the days of FTIR observations. These backward trajectories showed that nearly 30 % of airmasses above Rikubetsu on the FTIR observation days came from Chinese region as is shown in Figure 2-3. We add these descriptions in Section 5.2. There is no significant difference in the RMS residuals between baseline HFC-23 conditions and enhanced HFC-23 conditions. Mean RMS residual and its standard deviation for baseline condition (DJF) in 2006-2010 are 0.296±0.034, while those for enhanced condition (other months) are 0.296±0.048. There is temperature dependency on the PLL and this explain 5% difference in maximum as described in Section 5.3. We have checked some correlations. Figures 2-4 and 2-5 are the correlations between HFC-23 and $H_2O$, and HFC-23 and $HDO/H_2O$, for Rikubetsu in 2006-2010, respectively. You can find positive correlations, but note that $H_2O$, $HDO/H_2O$, and also the transport from Eurasia are all high in summer season.

[Figure]

Figure 2-4. HFC-23 v.s. $H_2O$ correlation         Figure 2-5. HFC-23 v.s. $HDO/H_2O$ correlation

Figures 2-6 and 2-7 are the correlations between HFC-23 and $H_2O$, and HFC-23 and $HDO/H_2O$, for Syowa in 2007-2016, respectively. Here, there is no significant correlations. These results indicate that the retrieval artifacts from $H_2O$, and $HDO/H_2O$ are not significant. We add description on this in Section 5.2.

[Figure]

[Figure]

Figure 2-6. HFC-23 v.s. H₂O correlation.     Figure 2-7. HFC-23 v.s. HDO/H₂O correlation

*<2-4> In my opinion, asserting that the HFC-23 enhancements were real would at minimum require comparing residuals for enhanced versus background conditions, ideally with a similar quality in both cases and no evidence of additional significant systematic features in the residuals for the enhanced case.*

As mentioned in the reply for the previous comment, there is no significant difference in the residuals between baseline HFC-23 conditions and enhanced HFC-23 conditions

*<2-5> The authors use two microwindows in the retrieval.   I assume independent 'background correction' parameters (slope and curvature) are used in each window.   Is it possible to have SFIT4 use a single set of background correction parameters spanning the two windows?   The windows are close together, so I expect a unified curve for the background correction should be viable.   This adjustment would presumably make the retrieval less susceptible to artifacts.*

Since we are using relatively wide two microwindows ranging 20 cm⁻¹ in total and 6 cm⁻¹ apart each other, we consider that using independent background correction parameters is better than using a single background correction parameter.

*<2-6> Page 8, line 25: for all observed spectra with the IFS-120/5HR instrument no ILS function was used because the instrument has always been maintained best optical alignment*
*    This statement is not entirely accurate.   An instrumental line shape function is still required, associated with the FTS scan length and the finite field of view.   It would be more appropriate*

*to say something like the ILS is accurately defined by the theoretical model for the given instrument configuration.*

You are right. We modified the description as you suggested.

*<2-7> Page 4, line 9: However, the ACE-FTS observations do not have sensitivity to the troposphere where all HFC-23 emissions occur.*

   *Also not entirely accurate as stated.   ACE-FTS measurements extend into the troposphere (i.e., below the tropopause). It would be more accurate to say ACE-FTS measurements do not extend low into the troposphere or do not probe near the surface.*

You are right. Also reviewer #3 mentioned on this by his comment <3-15>. We finally deleted the description of tropospheric sensitivity issue from the draft.

*<2-8> Page 20, line 3: a discussion of temperature sensitivity in relation to the ACE-FTS.*

   *Note that ACE-FTS analysis uses the cross sections directly in the analysis (via a bilinear interpolation in pressure and temperature), whereas the current study employs pseudolines derived from a set of cross sections.   I would assume that the deficiencies in temperature dependence described in the paper are a property of the pseudolines and not the cross sections themselves.   There could conceivably be temperature consistency issues arising from the ACE-FTS interpolation approach, but I see no reason to expect it is similar in nature to the pseudoline temperature dependence deficiencies unless one claims the temperature dependence problems are inherent to the Harrison cross sections (I would need to see proof, if that were claimed).*

You are right. We mixed up temperature dependence of pseudolines with that of cross sections. We deleted description of ACE-FTS issue here, and inserted it in the next paragraph.

*<2-9> Page 7, line 25: only about 1% of the atmospheric transmittance of solar infrared radiation at ground level (see Figure 3).*

   *It is traditional to make reference to figures in order.   Figure 3 is referenced before Figures 1 and 2.*

You are right. The reference for Figure 3 was deleted here.

*<2-10> Page 17, line 21: g is the column-averaged acceleration*

   *acceleration due to gravity*

It was corrected as suggested.

<2-11> Grammar/wording issues:
* * *
>Page 3, line 6: all UNFCCC CDM project were terminated ... [projects]

It was corrected as suggested.

>Page 5, line 31: detected with the MCT detector ... [measured with]

It was corrected as suggested.

>Page 5, line 31: Note, that ... [no comma]

It was corrected as suggested.

>Page 11, line 31: DOFS for HDO was increased when the wider window used ... [was used]

It was corrected as suggested.

>Page 14, line 7: due to the lack of vertically measurement data ... [vertically resolved]

It was corrected as suggested.

>Page 14, line 31: The SZA random uncertainty was assumed an uncertainty of 0.15° ... [assumed a value of]

It was corrected as suggested.

>Page 15, line 18: Since the MW for CH4 pre-retrieval is closed to the HFC-23 MWs ... [close to]

It was corrected as suggested.

>Page 16, line 29: These relative biases lead underestimation to the trend ... [lead to an

*underestimation of the trend]*

It was corrected as suggested.

*>Page 19, line 32: in all temperature region ... [regions]*

It was corrected as suggested.

*<2-15> In conclusion, while I think it is a good paper, I would like to see more said on AGAGE results and results from studies employing the Harrison cross sections (this study and ACE-FTS work) exhibiting a fairly consistent bias, suggesting a systematic error somewhere.   I would also like to see further proof that the apparent seasonal enhancement in HFC-23 is real (e.g., comparing residuals in the enhanced case versus the background case).*

Thank you. We added some more descriptions on the negative bias issue, including the ACE-FTS negative bias, in the conclusion. As is described in the response to your comment <2-3>, we believe that the seasonal enhancement of HFC-23 is real. The description of backward trajectory calculation was also added in the conclusion.

Ver. 4.8a, 2021/04/08(Thu) 16:00

[revised manuscript text omitted]

$$\boldsymbol{\varepsilon}_{\text{ret}} = \boldsymbol{A}_{\text{Tar,Int}}(\boldsymbol{x}_{\text{t}}^{\text{Int}} - \boldsymbol{x}_{\text{a}}^{\text{Int}}) + \boldsymbol{A}_{\text{Tar,Oth}}(\boldsymbol{x}_{\text{t}}^{\text{Oth}} - \boldsymbol{x}_{\text{a}}^{\text{Oth}}), \tag{13}$$

where $\boldsymbol{A}_{\text{Tar,Int}}$ is a part of the full averaging kernel matrix $\boldsymbol{A}$ where the row elements run over all target components and the column elements run over all interfering species; $\boldsymbol{A}_{\text{Tar,Oth}}$ is a part of the $\boldsymbol{A}$ matrix where the row and column elements run over all target and other parameter components, respectively; $\boldsymbol{x}_{\text{t}}^{\text{Int}}$ and $\boldsymbol{x}_{\text{a}}^{\text{Int}}$, $\boldsymbol{x}_{\text{t}}^{\text{Oth}}$ and $\boldsymbol{x}_{\text{a}}^{\text{Oth}}$ are the true and a priori state vectors of interfering species and other parameters, respectively. To estimate the retrieval errors from the interfering gases, the variabilities around the a priori profiles for $H_2O$ (HDO) were set to 10% and the uncertainties for other species were set to the values calculated from the appropriate WACCM datasets.

In order to estimate the non-retrieved forward model parameter error, the  covariance matrix $\boldsymbol{S}_{\text{f}}$ is calculated as:

$$\boldsymbol{S}_{\text{f}} = (\boldsymbol{GK}_{\text{b}})\boldsymbol{S}_{\text{b}}(\boldsymbol{GK}_{\text{b}})^{T}, \tag{14}$$

where $\boldsymbol{S}_{\text{b}}$ is the model parameter covariance matrix, which is derived from the uncertainties in the model parameters. For the random and systematic uncertainties of temperatures at Rikubetsu and Syowa Station, the uncertainties reported from the NCEP temperature profiles were assumed. The uncertainty of temperature at Rikubetsu is about 2 K in the troposphere, 2–10 K between the tropopause and 60 km, and 10 K above 60 km. The uncertainty of temperature at Syowa Station is about 2.5 K in the altitude range from the surface to 20 km, 2.5–10 K between 20 and 60 km, and 10 K above 60 km. The SZA random uncertainty was assumed to be of 0.15°, considering measurement time. For HFC-23, $N_2O$, $O_3$, $H_2O$, and HDO, the uncertainties of the spectroscopic parameters (i.e. line intensity, $S_v$; air-broadening coefficient, $\gamma_{\text{air}}$; coefficient of temperature dependence for $\gamma_{\text{air}}$, $n_{\text{air}}$) were also estimated. The uncertainties of $S_v$, $\gamma_{\text{air}}$, and $n_{\text{air}}$ of HFC-23, we set to 10%, 15%, and 15%, respectively, based on the PLL database (see https://mark4sun.jpl.nasa.gov/data/spec/Pseudo/CHF3_PLL_Update.pdf). For heavy molecules like HFC-23, ground state energy $E''$ values, which are relevant to the temperature dependency of $S_v$, are empirically given so that their uncertainties are larger than for light molecules (e.g., $H_2O$, $O_3$). In addition, the $E''$ uncertainty has a larger effect on $S_v$ at a cold site like Syowa Station. We assumed an error of 50 cm$^{-1}$ for the $E''$ values of the HFC-23 PLL, and estimated  uncertainties of 10% and 15% at Rikubetsu and Syowa Station, respectively, as the effect of the $E''$ error on $S_v$. For $N_2O$, $O_3$, $H_2O$ and HDO, the spectroscopic uncertainties were derived from the HITRAN 2008 database. The uncertainties for $N_2O$ and $O_3$ were set to 5%, 10%, and 5% for $S_v$, $\gamma_{\text{air}}$, and $n_{\text{air}}$, respectively. For $H_2O$ and HDO, we assigned an uncertainty of 10% to each parameter.

The measurement error was calculated from the error covariance matrix $\boldsymbol{S}_{\text{n}}$ defined as:

$$\boldsymbol{S}_{\text{n}} = \boldsymbol{G}\boldsymbol{S}_{\varepsilon}\boldsymbol{G}^{T}, \tag{15}$$

where $\boldsymbol{S}_\varepsilon$ is the measurement noise covariance matrix. We adopted the square inverse of the  SNR from the fitted residuals of the last iteration for the diagonal elements of $\boldsymbol{S}_\varepsilon$ as mentioned in Section 3.5.

Furthermore, we estimated the impact of the interfering $CH_4$ onto the HFC-23 retrieval because the retrieved HFC-23 total column is affected by the retrieval uncertainty of the pre-fitted $CH_4$ profile. The uncertainties of the pre-retrieved $CH_4$

5   total columns are dominated by the systematic uncertainties of its spectroscopic parameters. Considering the spectroscopic parameter uncertainty provided by the HITRAN2008 database, the mean uncertainties of $S_\nu$, $\gamma_{air}$, and $n_{air}$ on the pre-retrieved $CH_4$ total columns were approximately 5%, 4%, and 1%, respectively, at both sites. Since the MW for $CH_4$ pre-retrieval is close to the HFC-23 MWs, these spectroscopic uncertainties on $CH_4$ are partly cancelled between both MWs. Therefore, we assumed that the uncertainties of $S_\nu$, $\gamma_{air}$, and $n_{air}$ for $CH_4$ are 3%, 3%, and 1%, respectively, in the HFC-23

10  MWs. The effects of the $CH_4$ systematic uncertainties onto the retrieved HFC-23 total column were calculated from Equation (14) using these uncertainties. On the other hand, the effect of the $CH_4$ random uncertainty onto the retrieved HFC-23 was derived from the $1\sigma$ variability on the pre-retrieved $CH_4$ total columns. The $1\sigma$ standard deviations at Rikubetsu and Syowa Station were 4% and 3%, respectively. To quantity this uncertainty, we tested the HFC-23 retrievals by making the pre-retrieved $CH_4$ profiles scaled by $\pm4\%$ and $\pm3\%$ at Rikubetsu and Syowa Station, respectively. Then we calculated the percent

15  difference between the HFC-23 total column retrieved with the scaled $CH_4$ profile ("Scaled $CH_4$") and the ones retrieved with the no-scaled $CH_4$ profile ("Normal"). The percent difference $D$ is defined as:

$$D\,[\%]= \frac{TC_{\text{HFC-23,Scaled CH}_4}-TC_{\text{HFC-23,Normal}}}{(TC_{\text{HFC-23,Scaled CH}_4}+TC_{\text{HFC-23,Normal}})/2}\times100, \tag{16}$$

[revised manuscript text omitted]
| Rikubetsu (IFS-120/5HR) | 2019–2020 | 30 / 30 | 350 / 414 | $0.27 \pm 0.03$ | $1.0 \pm 0.01$ | $5.59 \pm 0.43$ |
| Syowa Station (IFS-120M) | 2007–2016 | 206 / 207 | 294 / 308 | $0.43 \pm 0.38$ | $1.0 \pm 0.03$ | $3.69 \pm 1.35$ |

**Table 4: Mean random and systematic errors and uncertainties on FTIR-retrieved HFC-23 total columns at Rikubetsu and Syowa Station.**

| Site (period) | Rikubetsu (1997–2010) | | Syowa Station (2007–2016) | |
|---|---|---|---|---|
| Error component | Random error [%] | Systematic error [%] | Random error [%] | Systematic error [%] |
| Smoothing | 1.4 | | 0.56 | |
| Retrieved parameters | 0.15 | | 0.070 | |
| Interfering species | 2.8 | | 0.51 | |
| Measurement | 12 | | 6.8 | |
| Temperature | 3.8 | 3.8 | 1.2 | 1.2 |
| SZA | 1.1 | | 2.5 | |
| $S_v$ of HFC-23 | | 10 | | 10 |
| $E''$ of HFC-23 | | 10 | | 15 |
| $\gamma_{air}$ of HFC-23 | | 3.8 | | 3.7 |
| $n_{air}$ of HFC-23 | | 0.51 | | 0.59 |
| $S_v$ of $N_2O$ | | 0.16 | | 0.072 |
| $\gamma_{air}$ of $N_2O$ | | 4.4 | | 1.3 |
| $n_{air}$ of $N_2O$ | | 0.79 | | 0.30 |
| $S_v$ of $O_3$ | | 0.063 | | 0.037 |
| $\gamma_{air}$ of $O_3$ | | 0.13 | | 0.088 |
| $n_{air}$ of $O_3$ | | 0.054 | | 0.038 |
| $S_v$ of $H_2O$ | | 0.048 | | 0.055 |
| $\gamma_{air}$ of $H_2O$ | | 6.6 | | 2.1 |
| $n_{air}$ of $H_2O$ | | 0.24 | | 0.13 |
| $S_v$ of HDO | | 0.070 | | 0.069 |
| $\gamma_{air}$ of HDO | | 15 | | 2.3 |
| $n_{air}$ of HDO | | 0.47 | | 0.15 |
| $CH_4$ pre-retrieved profile | 7.3 | | 4.4 | |
| $S_v$ of $CH_4$ | | 5.8 | | 4.4 |
| $\gamma_{air}$ of $CH_4$ | | 0.038 | | 0.063 |
| $n_{air}$ of $CH_4$ | | 0.012 | | 0.026 |
| Subtotal | 15 | 24 | 8.6 | 19 |
| Total | 28 | | 21 | |

| Site (period) | Rikubetsu (1997–2010) | | | Syowa Station (2007–2016) | | |
|---|---|---|---|---|---|---|
| Error component | Uncertainty | Random [%] | Systematic [%] | Uncertainty | Random [%] | Systematic [%] |
| Smoothing | [a] | 1.4 | | [a] | 0.56 | |
| Retrieved parameters | [a] | 0.15 | | [a] | 0.070 | |
| Interfering species | [a] | 2.8 | | [a] | 0.51 | |
| Measurement | [a] | 12 | | [a] | 6.8 | |
| Temperature | 2–10 K | 3.8 | 3.8 | 2.5–10 K | 1.2 | 1.2 |
| SZA | 0.15 ° | 1.1 | | 0.15 ° | 2.5 | |

| | | | | | |
|---|---|---|---|---|---|
| $S_\nu$ of HFC-23 | 10% | | 10 | 10% | 10 |
| $E''$ of HFC-23 | 10% | | 10 | 15% | 15 |
| $\gamma_{air}$ of HFC-23 | 15% | | 3.8 | 15% | 3.7 |
| $n_{air}$ of HFC-23 | 15% | | 0.51 | 15% | 0.59 |
| $S_\nu$ of $N_2O$ | 5% | | 0.16 | 5% | 0.072 |
| $\gamma_{air}$ of $N_2O$ | 5% | | 4.4 | 5% | 1.3 |
| $n_{air}$ of $N_2O$ | 10% | | 0.79 | 10% | 0.30 |
| $S_\nu$ of $O_3$ | 5% | | 0.063 | 5% | 0.037 |
| $\gamma_{air}$ of $O_3$ | 5% | | 0.13 | 5% | 0.088 |
| $n_{air}$ of $O_3$ | 10% | | 0.054 | 10% | 0.038 |
| $S_\nu$ of $H_2O$ | 10% | | 0.048 | 10% | 0.055 |
| $\gamma_{air}$ of $H_2O$ | 10% | | 6.6 | 10% | 2.1 |
| $n_{air}$ of $H_2O$ | 10% | | 0.24 | 10% | 0.13 |
| $S_\nu$ of HDO | 10% | | 0.070 | 10% | 0.069 |
| $\gamma_{air}$ of HDO | 10% | | 15 | 10% | 2.3 |
| $n_{air}$ of HDO | 10% | | 0.47 | 10% | 0.15 |
| $CH_4$ pre-retrieved profile | [a] | 7.3 | | [a] | 4.4 |
| $S_\nu$ of $CH_4$ | 3% | | 5.8 | 3% | 4.4 |
| $\gamma_{air}$ of $CH_4$ | 1% | | 0.038 | 1% | 0.063 |
| $n_{air}$ of $CH_4$ | 3% | | 0.012 | 3% | 0.026 |
| Subtotal | | 15 | 24 | 8.6 | 19 |
| Total | | 28 | | 21 | |

[a] These uncertainties are described in detail in Section 4.2.

**Table 5: HFC-23 annual growth rates and standard errors derived from monthly mean $X_{HFC-23}$ at Rikubetsu and Syowa Station, in ppt year⁻¹. The annual growth rates computed from the AGAGE annual global mean dataset, the CGAA air sample dataset, and the AGAGE in-situ measurements at THD and CGO are also listed for the same periods, unless indicated by other time frames lists in brackets.**

|  |  |  |  |  |
|---|---|---|---|---|
|  |  |  |  |  |
|  |  |  |  |  |
|  |  |  |  |  |
|  |  |  |  | |
|  |  |  |  |  |
|  |  |  |  |  |
|  |  |  |  |  |

| Observation Site / Dataset | Annual Change [ppt year⁻¹] | | |
|---|---|---|---|
|  Data Period | 1997–2009 | 2008–2019 | 2007–2016 |
|  |  |  | – |
| Rikubetsu DJF (FTIR) | 0.810 ± 0.093 | 0.928 ± 0.108 | – |
| Syowa Station (FTIR) | – | – | 0.823 ± 0.075 |
| Annual Gglobal Mmean (12-box model) | 0.820 ± 0.013 | 0.892 ± 0.030 (2008–2016) | 0.878 ± 0.020 |
| CGAA | 0.805 ± 0.006 (1997–2009) | – | – |
| THD (AGAGE in-situ) | – | 0.994 ± 0.001 (2007–2019) | – |
| CGO (AGAGE in-situ) | – |  | 0.874 ± 0.002 |

**Figures**

[Figure]

Figure 1:  Examples of solar absorption spectra taken from FTIR observations at Syowa Station.  The red spectrum was obtained with the filter #6 on 30 September 2007.  The green and the blue ones were measured with filter #7 and #8 on 30 September 2011, respectively.  A positive zero-level offset is clearly seen on the red filter #6 spectrum.

[Figure]

Figure 2: Time-series of the total columns of HFC-23 and CH$_4$ retrieved from FTIR infrared spectra observed at Syowa Station in 2007. (a) HFC-23 total columns (red-x plots) derived from HFC-23 retrievals accompanied by column-retrieval (scaling) of CH$_4$ profile, and the scaled CH$_4$ total columns (green-x plots). (b) The correlation between HFC-23 and CH$_4$ of (a). (c) Independent retrieved CH$_4$ total columns using a spectral region from 1201.820 to 1202.605 cm$^{-1}$ (green dots), and HFC-23 total columns from retrievals using the independent retrieved CH$_4$ profiles as fixed profiles (red dots). (d) The correlation between HFC-23 and CH$_4$ of (c). Note that these retrieved HFC-23 columns were selected by the threshold of the fitted RMS value depending on the value of solar zenith angle (SZA): the threshold of the fitted RMS are < 0.5% for SZA < 85º and < 1.5% for SZA of 85% or greater.

[Figure]

**Figure 3: Typical spectral simulation results of the two HFC-23 retrieval micro-windows (left panel: MW1; right panel: MW2) fitted to the observed spectrum at Syowa Station on 9 November 2011 at 13:47 UTC. The top two panels show the residuals (observed minus calculated) of the fittings for MW1 and MW2. The middle two panels show the absorption contributions of HFC-23, PAN, HCFC-141b, HCFC-142b, and HCFC-22 in MW1 and MW2. The bottom two panels show the individual contributions from each interfering species, shifted by multiples of 0.025 for clarity, except the observed and the calculated lines.**

[Figure]

**Figure 4:** **Typical averaging kernels of the HFC-23 retrieval for the same spectrum shown in Figure 3, which are normalized using the a priori profile. Note that the vertical scale is from surface up to 60 km because there is almost no sensitivity above 60 km.**

[Figure]

**Figure 5: (a): Time-series of FTIR-retrieved HFC-23 total columns with total random errors at Rikubetsu and Syowa Station. (b): The fitted RMS values on individual retrieved total column. The total columns at Rikubetsu and Syowa Station are shown by green circles and blue triangles, respectively. The fitted RMS values at Rikubetsu and Syowa Station are shown by circles and triangles, respectively, with the color-coding depended on the SZA.**

[Figure]

**Figure 6: The fitted RMS residuals versus the SZA values on individual retrieval. The RMS values at Rikubetsu and Syowa Station are shown by green circles and blue triangles, respectively.**

[Figure]

[Figure]

**Figure 7: Time-series of the monthly mean FTIR-retrieved $X_{\text{HFC-23}}$ at Rikubetsu and Syowa Station, along with the AGAGE in-situ measurements at CGO and THD, and the annual global mean mole fractions and the Cape Grim Air Archive samples, which were reported by Simmonds et al. (2018b).**

[Figure]

**Figure 8:** Seasonal cycles of the FTIR-retrieved $X_{\text{HFC-23}}$ at Rikubetsu for the 1997–2020 period and at Syowa Station for the 2007–2016 period.

[Figure]

**Figure 9:** **(a):** Retrieved mole fraction scaling factors from four HFC-23 laboratory spectrum datasets using the 2020 HFC-23 PLL at the spectral region from 1105–1240 cm$^{-1}$ plotted versus temperature. **(b):** The temperature and pressure conditions of the laboratory measurements of Harrison (2013).

---

## Author Comment (AC3)

**Reply to reviewer #3**

We thank anonymous reviewer #3 for his/her constructive review that would improve the contents of our paper. The review comments by anonymous reviewer #3 are numbered and repeated below as *in italic letters*, followed by our answers. In the new draft with corrections (supplement file), red, green, purple, and blue corrections are the revisions suggested by reviewers #1, #2, #3, and co-authors, respectively.

*<<Reviewer #3>>*

*<3-1> This paper describes the derivation of total column and dry-air mixing ratios of HFC-23, a compound of interest to the atmospheric science community because of its non-negligible contributions to radiative forcing, its long lifetime, and because controls exist on its emissions. Accurate retrievals of this chemical would be important for providing useful independent assessments of its atmospheric burden and how that has changed over time. It is clear that the challenge in providing accurate retrievals is a difficult one, given the very small absorbance that is involved, and because a number of other gases are potentially interfere. The authors have clearly considered many of the factors complicating this retrieval, but the manuscript could use some additional clarifications and considerations before publishing.*

Thank you. We try to make additional clarifications and considerations.

*<3-2> Suggesting that the apparent seasonality represents emissive influences is premature, in my opinion, without any discussions of sensitivities and likelihoods. Before this assertion seems at all possible, one would have to explore a number of things (inversion analysis not necessary):*

We try to add some additional discussion on the seasonality of the HFC-23 retrievals as follows.

*<3-3> Seasonal wind patterns reaching the site. Do they vary in a way that is potentially consistent with transport from potential source regions in Spring and not in the others? How about 2018-2019, where the seasonality seems much less pronounced. Given that emissions continue during these years, and perhaps predominantly from a region (China) close to Japan, one might expect larger seasonality in the more recent results, not reduced seasonality such as is observed.*

We calculated 10-days backward trajectories for all FTIR measurement days from Rikubetsu

originated at 2000 m. The attached Figures 3-1, 3-2, and 3-3 represents the 10-days backward trajectories for the FTIR measurement days in 2006, 2019, and 2020, respectively. We assumed that the rectangular area (20°N-45°N, 110°E-125°E) shown by red dotted lines in these figures represents the Chinese area, from where large HFC-23 emission is expected. As is shown in these figures, there are several trajectories which passed north part of Chinese area in 2006, while there in only one trajectory which passed Chinese area both in 2019 and 2020. The percentages of trajectories which passed Chinese area are: 29%, 5%, and 11%, respectively. This result explains why the analyzed HFC-23 don't show apparent seasonal variation in 2019 and 2020, when the FTIR observed only few airmass originated from China.

[Figure]

Figure 3-1. Trajectories in 2006

[Figure]

Figure 3-2. Trajectories in 2019

[Figure]

Figure 3-3. Trajectories in 2020

*<3-4> While it is very good to see the discussion on a factor changing seasonally that affects the retrieved information (temperature), but its influence is insufficient to explain the unusually high amplitude that there is no hint of in the surface data. This is very puzzling. Why, for example, is the seasonality in the NH result much less in 2019-2020? Is it because the newer instrument is less susceptible to interfering influences, so in fact not the result of such a large emission?*

As is explained in the reply to your comment <3-3> above, the backward 10-days trajectory in 2019 and 2020 did not often pass through Chinese area compared with 2006. We consider this is the major reason why we did not see large seasonality in 2019-2020.

*<3-5> The seasonal changes in total column and mole fraction are on the other of a factor of 2. If this were truly an emissive signal, the consistency with which is observed during spring would enhanced mole fractions over a large region of the mid-latitude NH throughout an entire season and, hence, very large emissions. What emission magnitude would this demand, and it is reasonable given the global emissions derived for these years? Some qualitative discussion of these possibilities is warranted to determine if the hypothesis is, or is not, reasonable.*

Although we are not quite familiar with the HFC-23 emission magnitude from Chinese area, we could see apparent differences among the spikes of surface HFC-23 measurements by AGAGE. The following three figures represents the continuous surface AGAGE measurement of HFC-23 at Fig. 3-4: Hateruma (HAT), Fig. 3-5: Cape Ochiishi (COI), and Fig. 3-6: Trinidad Head (THD). Since Hateruma station is quite close to China (about 500 km apart), it often captures the HFC-23-rich airmasses which is shown by the spiky dots in Fig. 3-4. Cape Ochiishi (43.2°N, 145.5°E), which is located rather close (about 150 km) to Rikubetsu where our FTIR is located, is about 2,000 km apart from China. Nevertheless, HFC-23 data in COI sometimes show spikes as shown in Fig. 3-5, which suggests medium-range transport of HFC-23 rich air-masses from China. The HFC-23 data in Trinidad Head (>10,000 km apart from China) seldom show spikes, which means that HFC-23 rich air was almost diluted after inter-continental transport across the Pacific.

[Figure]

Fig. 3-4 HFC-23 measurement at Hateruma

[Figure]

Fig. 3-6 HFC-23 measurement at Trinidad Head

[Figure]

Fig. 3-5 HFC-23 measurement at Ochiishi

*<3-6> On trends. It is not clear to the reader why it is important to express and assess trends for multiple periods and some further explanation on this point is needed, especially because it isn't apparent that there is a change in atmospheric growth rates corresponding to the chosen dates. Is it because of concurrent changes in instrumentation? Is it relevant to be deriving trends for significantly modified instruments? Especially over a span of time during which very few retrievals were made? The main conclusion is that this methodology provides accurate tracking of HFC-23 atmospheric mole fraction trends, yet there are significant differences in results and trends that aren't well caveated in this main conclusion.*

The selection of trend period was not appropriate in the previous draft. We modified the selection of trend period more appropriately to represent the period when data were available. The Table 5 was modified to show three new periods. The contents in Section 5.4 was also modified. Now, the derived trends agreed within their standard deviation.

*Other details:*

*<3-7> 19, line 14. This text is missing DJF, I believe: "indicated in Figure 8 that the retrievals at Rikubetsu have a negative bias".*

It is not appropriate to refer Figure 8 here, so it was deleted. The term 'DJF' was added as suggested.

*<3-8> An indication of the accuracy for the CH4 pre-retrieval isn't provided, but would be useful to understand to know if it is accurate or has biases that might affect the HFC-23 retrieval.*

The accuracy of the $CH_4$ pre-retrieval is written in the paragraph starting page 15, line 14. Since we are using relative absorption differences of $CH_4$ lines for the pre-retrieval and for the $CH_4$ line overlapped with HFC-23, their absolute accuracy or bias will not affect the HFC-23 retrieval.

*<3-9> 12, lines 3-22 needs to more clearly written. Explain at first the different approaches that are available, and the terminology, so that the average reader will understand.*

We modified the description of this paragraph to explain why we introduced '$CH_4$ pre-retrieval' more clearly, at the beginning of this paragraph.

*<3-10> Consider subsetting data in some figures in an additional panel to allow the reader a better view of results that are relevant. Make clear, if true, that results in Figure 7 only include retained data with acceptable RMS values.*

We agree that old Figure 7 is too small and difficult to see individual points. We extended the vertical size of new Figure 7. Yes, the results in Figure 7 only include the retained data with acceptable RMS values as is describe in Section 5.1.1.

*<3-11> Abstract describes the work quite well, although more specificity and clarity is needed in lines 29-33. What is the size of the negative biases? Why are results from only some months used in the NH? On p.2, lines 4-5, having the capability to measure trends begs the long-term difference in NH results, [MORE HERE??]*

We added the description of amount of negative biases and their possible cause due to the HFC-23 spectroscopy in the abstract. The reason of the selection of DJF data at Rikubetsu is also

described. The advantage of addition of FTIR long-term measurement to fill the spatial and temporal gaps of the AGAGE observation is also added at the last of the abstract.

*<3-12> 2, line 22. Increasing emissions of HCFC-22 do not necessarily mean increasing emissions of HFC-23, given that HFC-23 is associated with the production of 22, not its emission, which is delayed by use in appliances. Focus on the tie between HCFC-22 production with HFC-23 emissions.*

We agree to the reviewer that HFC-23 increase is associated with the production of HCFC-22, not its emission. We deleted the word 'Clearly' here.

*<3-13> 2, line 23-24. ODPs have been calculated for HFCs, so they are in fact non-zero. The destruction arises from changes in the thermal structure of the atmosphere. Consider rephrasing "because they do not contain ozone depleting halogen atoms".*

We agree. We modified the text as suggested.

*<3-14> p.3, lines 12-13. A citation is needed here. Are these truly mandated? Or are they aspirational goals to reduce emissions. Have India or China ratified the Kigali Amendment?*

There was a misunderstanding. India and China have not ratified the Kigali Amendment yet. We deleted the description of China and India from the draft.

*<3-15> 4, lines 8-9. This point does not seem all that relevant given the HFC-23 is a long-lived gas who's mole fraction is fairly evenly distributed.*

We agree to the reviewer. We deleted the description of tropospheric sensitivity issue from the draft.

Ver. 4.8a, 2021/04/08(Thu) 16:00

[revised manuscript text omitted]
| Rikubetsu (IFS-120/5HR) | 2019–2020 | 30 / 30 | 350 / 414 | $0.27 \pm 0.03$ | $1.0 \pm 0.01$ | $5.59 \pm 0.43$ |
| Syowa Station (IFS-120M) | 2007–2016 | 206 / 207 | 294 / 308 | $0.43 \pm 0.38$ | $1.0 \pm 0.03$ | $3.69 \pm 1.35$ |

**Table 4: Mean random and systematic errors and uncertainties on FTIR-retrieved HFC-23 total columns at Rikubetsu and Syowa Station.**

| Site (period) | Rikubetsu (1997–2010) | | Syowa Station (2007–2016) | |
|---|---|---|---|---|
| Error component | Random error [%] | Systematic error [%] | Random error [%] | Systematic error [%] |
| Smoothing | 1.4 | | 0.56 | |
| Retrieved parameters | 0.15 | | 0.070 | |
| Interfering species | 2.8 | | 0.51 | |
| Measurement | 12 | | 6.8 | |
| Temperature | 3.8 | 3.8 | 1.2 | 1.2 |
| SZA | 1.1 | | 2.5 | |
| $S_v$ of HFC-23 | | 10 | | 10 |
| $E''$ of HFC-23 | | 10 | | 15 |
| $\gamma_{air}$ of HFC-23 | | 3.8 | | 3.7 |
| $n_{air}$ of HFC-23 | | 0.51 | | 0.59 |
| $S_v$ of $N_2O$ | | 0.16 | | 0.072 |
| $\gamma_{air}$ of $N_2O$ | | 4.4 | | 1.3 |
| $n_{air}$ of $N_2O$ | | 0.79 | | 0.30 |
| $S_v$ of $O_3$ | | 0.063 | | 0.037 |
| $\gamma_{air}$ of $O_3$ | | 0.13 | | 0.088 |
| $n_{air}$ of $O_3$ | | 0.054 | | 0.038 |
| $S_v$ of $H_2O$ | | 0.048 | | 0.055 |
| $\gamma_{air}$ of $H_2O$ | | 6.6 | | 2.1 |
| $n_{air}$ of $H_2O$ | | 0.24 | | 0.13 |
| $S_v$ of HDO | | 0.070 | | 0.069 |
| $\gamma_{air}$ of HDO | | 15 | | 2.3 |
| $n_{air}$ of HDO | | 0.47 | | 0.15 |
| $CH_4$ pre-retrieved profile | 7.3 | | 4.4 | |
| $S_v$ of $CH_4$ | | 5.8 | | 4.4 |
| $\gamma_{air}$ of $CH_4$ | | 0.038 | | 0.063 |
| $n_{air}$ of $CH_4$ | | 0.012 | | 0.026 |
| Subtotal | 15 | 24 | 8.6 | 19 |
| Total | 28 | | 21 | |

| Site (period) | Rikubetsu (1997–2010) | | | Syowa Station (2007–2016) | | |
|---|---|---|---|---|---|---|
| Error component | Uncertainty | Random [%] | Systematic [%] | Uncertainty | Random [%] | Systematic [%] |
| Smoothing | [a] | 1.4 | | [a] | 0.56 | |
| Retrieved parameters | [a] | 0.15 | | [a] | 0.070 | |
| Interfering species | [a] | 2.8 | | [a] | 0.51 | |
| Measurement | [a] | 12 | | [a] | 6.8 | |
| Temperature | 2–10 K | 3.8 | 3.8 | 2.5–10 K | 1.2 | 1.2 |
| SZA | 0.15 ° | 1.1 | | 0.15 ° | 2.5 | |

| | | | | |
|---|---|---|---|---|
| $S_\nu$ of HFC-23 | 10% | 10 | 10% | 10 |
| $E''$ of HFC-23 | 10% | 10 | 15% | 15 |
| $\gamma_{air}$ of HFC-23 | 15% | 3.8 | 15% | 3.7 |
| $n_{air}$ of HFC-23 | 15% | 0.51 | 15% | 0.59 |
| $S_\nu$ of $N_2O$ | 5% | 0.16 | 5% | 0.072 |
| $\gamma_{air}$ of $N_2O$ | 5% | 4.4 | 5% | 1.3 |
| $n_{air}$ of $N_2O$ | 10% | 0.79 | 10% | 0.30 |
| $S_\nu$ of $O_3$ | 5% | 0.063 | 5% | 0.037 |
| $\gamma_{air}$ of $O_3$ | 5% | 0.13 | 5% | 0.088 |
| $n_{air}$ of $O_3$ | 10% | 0.054 | 10% | 0.038 |
| $S_\nu$ of $H_2O$ | 10% | 0.048 | 10% | 0.055 |
| $\gamma_{air}$ of $H_2O$ | 10% | 6.6 | 10% | 2.1 |
| $n_{air}$ of $H_2O$ | 10% | 0.24 | 10% | 0.13 |
| $S_\nu$ of HDO | 10% | 0.070 | 10% | 0.069 |
| $\gamma_{air}$ of HDO | 10% | 15 | 10% | 2.3 |
| $n_{air}$ of HDO | 10% | 0.47 | 10% | 0.15 |
| $CH_4$ pre-retrieved profile | [a] | 7.3 | [a] | 4.4 |
| $S_\nu$ of $CH_4$ | 3% | 5.8 | 3% | 4.4 |
| $\gamma_{air}$ of $CH_4$ | 1% | 0.038 | 1% | 0.063 |
| $n_{air}$ of $CH_4$ | 3% | 0.012 | 3% | 0.026 |
| Subtotal | | 15 | 24 | 8.6 | 19 |
| Total | | 28 | | 21 |

[a] These uncertainties are described in detail in Section 4.2.

**Table 5:** HFC-23 annual growth rates and standard errors derived from monthly mean $X_{HFC-23}$ at Rikubetsu and Syowa Station, in ppt year$^{-1}$. The annual growth rates computed from the AGAGE annual global mean dataset, the CGAA air sample dataset, and the AGAGE in-situ measurements at THD and CGO are also listed for the same periods, unless indicated by other time frames lists in brackets.

|  |  |  |  |  |
|---|---|---|---|---|
|  |  |  |  |  |
|  |  |  |  |  |
|  |  |  |  |  |
|  |  |  |  | |
|  |  |  |  |  |
|  |  |  |  |  |
|  |  |  |  |  |

| Observation Site / Dataset | Annual Change [ppt year$^{-1}$] | | |
|---|---|---|---|
| Data Period | 1997–2009 | 2008–2019 | 2007–2016 |
|  |  |  | – |
| Rikubetsu DJF (FTIR) | 0.8107 ± 0.09387 | 0.928894 ± 0.108 | – |
| Syowa Station (FTIR) | – | – | 0.823 ± 0.075 |
| Annual Global Mmean (12-box model) | 0.820 ± 0.0131 | 0.89278 ± 0.0230 (2007–2016) | 0.878 ± 0.020 |
| CGAA | 0.805 ± 0.006 (1997–2009) | – | – |
| THD (AGAGE in-situ) | – | 0.99484 ± 0.0012 (2007–2019) | – |
| CGO (AGAGE in-situ) | – |  | 0.874 ± 0.002 |

**Figures**

[Figure]

**Figure 1: Examples of solar absorption spectra taken from FTIR observations at Syowa Station. The red spectrum was obtained with the filter #6 on 30 September 2007. The green and the blue ones were measured with filter #7 and #8 on 30 September 2011, respectively. A positive zero-level offset is clearly seen on the red filter #6 spectrum.**

[Figure]

Figure 2: Time-series of the total columns of HFC-23 and CH₄ retrieved from FTIR infrared spectra observed at Syowa Station in 2007. (a) HFC-23 total columns (red-x plots) derived from HFC-23 retrievals accompanied by column-retrieval (scaling) of CH₄ profile, and the scaled CH₄ total columns (green-x plots). (b) The correlation between HFC-23 and CH₄ of (a). (c) Independent retrieved CH₄ total columns using a spectral region from 1201.820 to 1202.605 cm⁻¹ (green dots), and HFC-23 total columns from retrievals using the independent retrieved CH₄ profiles as fixed profiles (red dots). (d) The correlation between HFC-23 and CH₄ of (c). Note that these retrieved HFC-23 columns were selected by the threshold of the fitted RMS value depending on the value of solar zenith angle (SZA): the threshold of the fitted RMS are < 0.5% for SZA < 85º and < 1.5% for SZA of 85% or greater.

[Figure]

**Figure 3: Typical spectral simulation results of the two HFC-23 retrieval micro-windows (left panel: MW1; right panel: MW2) fitted to the observed spectrum at Syowa Station on 9 November 2011 at 13:47 UTC. The top two panels show the residuals (observed minus calculated) of the fittings for MW1 and MW2. The middle two panels show the absorption contributions of HFC-23, PAN, HCFC-141b, HCFC-142b, and HCFC-22 in MW1 and MW2. The bottom two panels show the individual contributions from each interfering species, shifted by multiples of 0.025 for clarity, except the observed and the calculated lines.**

[Figure]

**Figure 4: Typical averaging kernels of the HFC-23 retrieval for the same spectrum shown in Figure 3, which are normalized using the a priori profile. Note that the vertical scale is from surface up to 60 km because there is almost no sensitivity above 60 km.**

[Figure]

**Figure 5: (a): Time-series of FTIR-retrieved HFC-23 total columns with total random errors at Rikubetsu and Syowa Station. (b): The fitted RMS values on individual retrieved total column. The total columns at Rikubetsu and Syowa Station are shown by green circles and blue triangles, respectively. The fitted RMS values at Rikubetsu and Syowa Station are shown by circles and triangles, respectively, with the color-coding depended on the SZA.**

[Figure]

**Figure 6: The fitted RMS residuals versus the SZA values on individual retrieval. The RMS values at Rikubetsu and Syowa Station are shown by green circles and blue triangles, respectively.**

[Figure]

[Figure]

**Figure 7:** Time-series of the monthly mean FTIR-retrieved $X_{\text{HFC-23}}$ at Rikubetsu and Syowa Station, along with the AGAGE in-situ measurements at CGO and THD, and the annual global mean mole fractions and the Cape Grim Air Archive samples, which were reported by Simmonds et al. (2018b).

[Figure]

**Figure 8:** Seasonal cycles of the FTIR-retrieved $X_{HFC-23}$ at Rikubetsu for the 1997–2020 period and at Syowa Station for the 2007–2016 period.

[Figure]

**Figure 9:** (a): Retrieved mole fraction scaling factors from four HFC-23 laboratory spectrum datasets using the 2020 HFC-23 PLL at the spectral region from 1105–1240 cm$^{-1}$ plotted versus temperature. (b): The temperature and pressure conditions of the laboratory measurements of Harrison (2013).

---

## Author Response (AR2)

**Reply to reviewer #3**

We thank anonymous reviewer #3 for his/her additional comments on our revised draft. The review comments by anonymous reviewer #3 are numbered and repeated below *in italic letters*, followed by our answers. In the re-revised draft with corrections (supplement file), red text are the revisions suggested by reviewers #3.

*<<Reviewer #3>>*

*<3-1> The authors have been very responsive to the reviewers' concerns in preparing their revised manuscript. 1 have only one additional issue that 1 still think requires attention:*

Thank you for your very positive assessment of our response.

*<3-2> Discussion of seasonality still needs improving to have it more accurately reflect what is presented in Figure 8. Figure 8 shows seasonal changes at Rik that are hardly outside the uncertainty bars displayed, and the discussion at one point focuses on substantial increases from March to May that aren't outside the uncertainties displayed in Figure 8. Results for only a few months (Dec,Jan,Feb) appear outside 1-sigma s.d. This then leads to the extended discussion of the possibility that emissions (and their transport) are responsible for the summertime elevations.*

Perhaps we need to clarify the meaning of the 1-simga s.d. in Figure 8. The 1-sigma s.d. in Figure 8 represents the year-to-year variability for each month, not the uncertainty of the measurements. The text was modified to "with large year-to-year variability (+/- 15-20%) for each month" to more clearly explain that the 1-sigma s.d. represent the variability among different years, not the measurement uncertainties. Although the standard deviations of mean values overlap with zero line in many months in Figure 8, the standard errors of mean values are much smaller by a factor of 3 to 4. It indicates the mean seasonal cycle is outside the uncertainties. We also modified the sentence for the cause of the spring-summer peak to "the peaks at Rikubetsu during spring-summer were affected by enhancements due to atmospheric transport from a region emitting HFC-23".

*<3-3> Results at Ochiishi are only somewhat useful to the reader at this point, but could be made much more informative. This is because they haven't been presented with respect to the seasonality they suggest. Why not a simple analysis of the seasonality there to see if it approaches +/-10%,*

*as may be suggested at Rik if uncertainties in retrievals there are ignored? If Ochiishi ground-based results don't show a seasonality like is observed in the total column, the authors could still argue that perhaps Ochiishi isn't seeing the total column incursion of emissions from the east.*

We analyzed hourly Ochiishi HFC-23 data for the open data period between 2006 and 2010, and took a detrended monthly mean, which is shown in Figure 3-1 below. Each year's data are plotted in dotted lines, while the average is thick green line with $1\sigma$ standard deviation in broken purple line. As you notice, the ground-based Ochiishi data also show peaks during spring-summer months in May, June, and July. This supports our FTIR analysis which has elevated values between April to July as shown in Figure 8.

[Figure]

Figure 3-1. Detrended monthly mean HFC-23 data at Ochiishi between 2006 and 2010.

*<3-4> At Syowa, it is argued fairly strongly that seasonality is not present. In retrieved info, despite a mean seasonal variability apparent in Figure 8 that is quite similar to Rik in most months, and visible uncertainties that appear to be smaller than at Rik, on average. But at Syowa in Figure 8, results aren't shown for all months (Jan, Feb, Jun appear to be missing), and uncertainties are not apparent for many other months. Perhaps there aren't enough data to describe the seasonality at Syowa? Some clarifications and reconsiderations are warranted.*

We added January 2008 HFC-23 data (which was not used in the analysis in the previous draft due to a technical issue) for Syowa Station in the analysis, and Figures 2, 5, 6, 7, 8 were modified. In the new Figure 8, data for February and June are still missing, and there are no 1-sigma s.d.

bars in January, May, July, and August because there is only one year monthly average data available. Nevertheless, seasonal variation is not so apparent in the case of Syowa Station. This point is modified in the text in Section 5.2 to "At Syowa Station, where there are no observations in February and June and only one year of observations for January, May, July, and August, …". We also improved the caption of Figure 8 accordingly.